# Chronology of motor-mediated microtubule streaming

**Arvind Ravichandran, Özer Duman, Masoud Hoore, Guglielmo Saggiorato[†], Gerard A Vliegenthart\*, Thorsten Auth\*, Gerhard Gompper\***

Theoretical Soft Matter and Biophysics, Institute of Complex Systems and Institute for Advanced Simulation, Forschungszentrum Jülich, Jülich, Germany

**Abstract** We introduce a filament-based simulation model for coarse-grained, effective motor-mediated interaction between microtubule pairs to study the time-scales that compose cytoplasmic streaming. We characterise microtubule dynamics in two-dimensional systems by chronologically arranging five distinct processes of varying duration that make up streaming, from microtubule pairs to collective dynamics. The structures found were polarity sorted due to the propulsion of antialigned microtubules. This also gave rise to the formation of large polar-aligned domains, and streaming at the domain boundaries. Correlation functions, mean squared displacements, and velocity distributions reveal a cascade of processes ultimately leading to microtubule streaming and advection, spanning multiple microtubule lengths. The characteristic times for the processes extend over three orders of magnitude from fast single-microtubule processes to slow collective processes. Our approach can be used to directly test the importance of molecular components, such as motors and crosslinking proteins between microtubules, on the collective dynamics at cellular scale.

DOI: https://doi.org/10.7554/eLife.39694.001

**\*For correspondence:**
g.vliegenthart@fz-juelich.de (GV);
t.auth@fz-juelich.de (TA);
g.gompper@fz-juelich.de (GG)

**Present address:** [†]LPTMS, CNRS, Université Paris-Sud, Université Paris-Saclay, Orsay, France

**Competing interests:** The authors declare that no competing interests exist.

## Introduction

The vigorous motion of the intracellular fluid, known as cytoplasmic streaming, is caused by cytoskeletal filaments and molecular motors. In *Drosophila* oocytes this cellular-scale fluid motion, which occurs over multiple time scales, is responsible for efficient mixing of ooplasm and nurse-cell cytoplasm, for long-distance transport of intracellular material, and for proper patterning of the oocyte (*Quinlan, 2016*; *Palacios and St Johnston, 2002*; *Lu et al., 2016*). Although cytoplasmic streaming is known for centuries (*Berthold, 1886*) and kinesin-1 molecular motors and microtubules (MTs) have been identified as the components responsible for ooplasmic streaming (*Quinlan, 2016*; *Palacios and St Johnston, 2002*; *Gutzeit, 1986*; *Serbus et al., 2005*), there is considerable debate about the aetiological mechanisms for force generation. Namely, the constituent events, their order of occurrences, and their characteristic durations, which ultimately give streaming are not understood. Some studies suggest that streaming is caused by the hydrodynamic entrainment of motor-transported cargos (*Monteith et al., 2016*; *Theurkauf et al., 1992*), others that it is due to the motor-mediated sliding of adjacent MTs (*Jolly et al., 2010*; *Lu et al., 2016*; *Winding et al., 2016*).

Motor-mediated MT sliding occurs because molecular motors crosslink adjacent MTs and use ATP (adenosine triphosphate) molecules as fuel to 'walk' on them unidirectionally in the direction of MT polarity (*Vale and Milligan, 2000*). This leads to significantly different active dynamics of MT pairs that are polar-aligned and antialigned (*Gao et al., 2015*; *Blackwell et al., 2016*; *Ravichandran et al., 2017*): Motors that crosslink polar-aligned MTs hold the polar-aligned MTs together, generating an effective attraction (*Ravichandran et al., 2017*). Active motors crosslink and slide antialigned MTs. The motors thus act as force dipoles that break nematic symmetry in MT solutions. In the absence of permanent crosslinkers, which are known to render the active network

contractile (*Belmonte et al., 2017*), this ultimately can cause large-scale flows in the cytoskeleton (*Jolly et al., 2010*; *Lu et al., 2016*). Several approaches to analyse the collective motion, such as the displacement correlation function or the analysis of velocity distributions, are inspired by studies on collective motion of self-propelled agents (*Duman et al., 2018*; *Needleman and Dogic, 2017*; *Zöttl and Stark, 2016*; *Bechinger et al., 2016*; *Elgeti et al., 2015*).

In vivo, individual MTs are stationary most of the time before suddenly undergoing a burst of long-distance travel with velocities reaching $\approx 10$ µm/s (*Jolly et al., 2010*). Also, fluorescence microscopy has shown the formation of long extended arms for an initially circular photoconverted area (*Jolly et al., 2010*). A possible mechanism for such a behaviour is illustrated in *Figure 1*: In the absence of active motor stresses, MTs in polar-aligned bundles diffuse slowly. When they encounter antialigned MTs, they can be actively and rapidly transported away from a polar-aligned bundle to another polar-aligned bundle where they again exhibit slow diffusive behaviour. These transitions between slow-bundling motion and fast-streaming bursts can give rise to Levy flight-like MT dynamics (*Chen et al., 2015*).

In vitro and in silico, MT-motor model systems allow a systematical study of the interactions between the basic components of the cytoskeleton. The systems often contain only a small number of different components and are usually also restricted in other respects, such as a reduced dimensionality or a lack of polymerization and depolymerization of the filaments. For example, a very well studied model system contains MTs and kinesin complexes at an oil-water interface (*Sanchez et al., 2012*; *DeCamp et al., 2015*), where poly(ethylene glycol)-induced depletion interaction both keeps the MTs at the interface and induces bundle formation. In contrast, coarse-grained computer simulations in 2D are used to study both structure formation on the bundle scale (*DeCamp et al., 2015*) as well as single-filament dynamics, for example to quantify the effect of motor properties or of presence of passive crosslinkers (*Ravichandran et al., 2017*; *Blackwell et al., 2016*; *Gao et al., 2015*; *Belmonte et al., 2017*). Because such model systems are simpler than the cytoskeleton in biological cells, they are especially suited to study specific mechanisms in detail.

Cytoplasmic streaming is a complex multi-scale phenomenon that cannot be fully understood using antialigned filaments alone. The importance of a specific mechanism can only be studied in vivo for a specific system. This so-called 'top-down' approach has been remarkably successful in describing streaming in the aquatic alga *Chara coranilla* (*Woodhouse and Goldstein, 2013*) and *Dropsophila* oocytes (*Khuc Trong et al., 2015*). In the former case, theoretical models have showed the importance of coupling hydrodynamic entrainment and microfilament dynamics to capture pattern formation relevant for streaming. In the latter case, simulations mimicking streaming in *Drosophila* oocytes have emphasised the importance of cortical MT nucleation in anteroposterior axis definition. It was shown that nucleation of MTs from the periphery is important to induce cytoplasmic flow patterns and to localise mRNAs in specific areas of the cell.

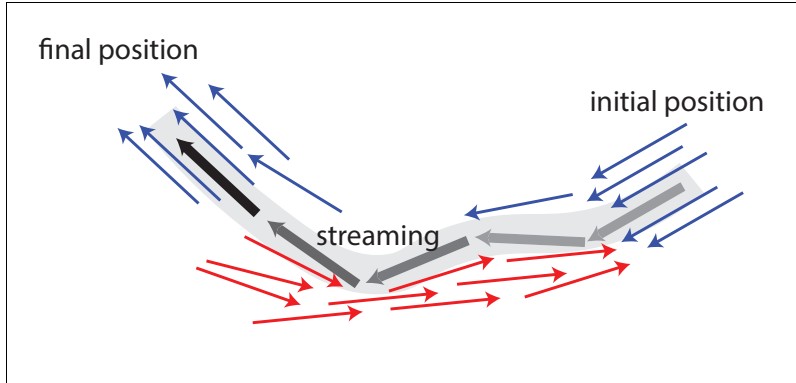

**Figure 1.** Schematic illustrating MT bundling and streaming. Polar-aligned MTs are coloured blue, and antialigned MTs are coloured red. The grey/black MT is transported from its initial position (grey), in one polar-aligned bundle, to its final position (black), to another polar-aligned bundle, via a stream.
DOI: https://doi.org/10.7554/eLife.39694.002

Microtubule-motor systems are intrinsically out of equilibrium, which has been shown for example by monitoring the dynamics of motors that walk along MTs (*Schnitzer and Block, 1997*) and by the violation of the fluctuation-dissipation theorem for tracer particles embedded into acto-myosin systems (*Mizuno et al., 2007*). Therefore, simulation approaches for systems of passive MTs at equilibrium have to be augmented with motor activity. Brownian dynamics simulations with MTs and explicit motors have been used to study network contractility (*Belmonte et al., 2017*), polarity-sorting, and stress generation at high MT densities (*Gao et al., 2015*), persistent motion of active vortices in confinement (*Head et al., 2011a*) and anomalous transport in active gels (*Head et al., 2011b*). Recent simulations for the defect dynamics in extensile MT systems have been performed on the coarse-grained level of MT bundles (*DeCamp et al., 2015*).

We employ a 'bottom-up' approach, where we study MT streaming induced by MT sliding using a model system. In order to characterize the dynamics in the system, we use a coarse-grained model to investigate whether a purely polarity-dependent MT-MT sliding mechanism, in the absence of any hydrodynamic forces, can be sufficient to capture large-scale streaming in bulk. We identify five distinct processes that comprise streaming with their characteristic times for various MT activities and surface fractions: (1) motor-driven MT sliding, (2) polarity-inversion, (3) maximal activity, (4) collective migration, and (5) rotation. Although various experimental studies have provided high spatial resolution to describe streaming phenomena (*Quinlan, 2016*), MT dynamics for streaming is still poorly understood. Inspired by the biological mechanism for MT-MT sliding, we use computer simulations to provide a novel temporal perspective into streaming for a wide range of time scales, which has not been achieved so far due to limitations of experimental techniques.

In order to capture cellular-scale dynamics in computer simulations, modelling individual motors along with MTs, although done before in several studies (*Mizuno et al., 2007*; *Hiraiwa and Salbreux, 2016*; *Ronceray et al., 2016*; *Head et al., 2014*; *Blackwell et al., 2016*; *Gao et al., 2015*; *Freedman et al., 2017*), can prove to be unwieldy due to the wide ranges of length and time scales involved. The sizes of individual kinesin molecules that crosslink and slide MTs are three orders of magnitude smaller than that of the cells within which they bring about large-scale dynamics. Also, there is a large disparity between the residence time of a cross-linking motor (10 seconds) (*Toprak et al., 2009*), and the characteristic time scale of motor-induced MT streaming or pattern formation in active gels (1 hr) (*Ganguly et al., 2012*; *Jolly et al., 2010*; *Lu et al., 2016*; *DeCamp et al., 2015*). In order to capture motor-induced cellular-level phenomena, such as organelle distribution, cytoplasmic streaming, and active cytoskeleton-induced lipid bilayer fluctuations, a coarse-grained description of cytoskeletal activity seems therefore appropriate.

## Coarse-grained model

Coarse-grained and continuum approaches are successfully applied to study cytoskeletal-motor systems. A well-developed model and simulation package is Cytosim that can be used to simulate flexible filaments together with further building blocks that, for example, act as nucleation sites, bind filaments together, and induce motility or severing (*Nedelec and Foethke, 2007*). It has been applied to study–among other processes in the cell–meiosis (*Burdyniuk et al., 2018*), mitosis (*Lacroix et al., 2018*), and centrosome centering (*Letort et al., 2016*). A different model that includes MT flexibility, MT polymerization and depolymerization, explicit motors, and hydrodynamics has recently been applied to study mitosis (*Nazockdast et al., 2017a*; *Nazockdast et al., 2017b*). Because the MTs are mostly radially oriented, steric interactions between MTs can be neglected and have not been taken into account. However, MT-MT repulsion is important to obtain nematic order at high MT densities, an essential ingredient for the bottom-up model systems containing suspensions of MTs and kinesins (*Sanchez et al., 2012*). Our model includes MT flexibility, effective-motor potentials, and excluded-volume interactions. Polymerization and depolymerization does not occur in the model system and is not taken into account. Effective-motor models in general aim to reduce the computational effort to efficiently study large systems (*Swaminathan et al., 2010*; *Jia et al., 2008*). Including hydrodynamic interactions using a particle-based approach is straightforward (*Winkler and Gompper, 2018*; *Müller et al., 2015*; *Gompper et al., 2009*), but beyond the scope of this paper.

## Microtubules

In our two-dimensional model, MTs are modelled as impenetrable, semi-flexible filaments of length $L$, thickness $\sigma$, and aspect ratio $L/\sigma$. Each of the $N$ filaments in the system is discretised into a chain of $n$ beads with diameter $\sigma$ that are connected by harmonic bonds. The configurational potential,

$$U_i = U_{\text{bond}} + U_{\text{angle}} + U_{\text{wca}}.$$ (1)

is the sum of passive potentials, that is the spring potential $U_{\text{bond}}$ between adjacent beads, the angle potential $U_{\text{angle}}$ between adjacent bonds, and the volume exclusion $U_{\text{WCA}}$ between MTs.

The bond energy,

$$U_{\text{bond}} = \frac{k_{\text{s}}}{2} \sum_{\text{bonds}} (r_{i,i+1} - r_0)^2,$$ (2)

acts between adjacent beads of the same MT. Here, $k_{\text{s}}$ is the bond stiffness, $r_0 = \sigma/2$ is the equilibrium bond length, and $r_{i,i+1} = |\mathbf{r}_{i,i+1}|$ is the distance between adjacent beads $i$ and $i+1$, which make up the MT. $U_{\text{angle}}$ is the bending energy, which is calculated using the position of three adjacent beads,

$$U_{\text{angle}} = \frac{\kappa}{r_0} \sum_{\text{angles}} (1 - \cos\theta_i),$$ (3)

that make up the angle $\theta_i$ (**Isele-Holder et al., 2016**). It acts between all groups of three adjacent beads that make up the same MT. The bending modulus $\kappa$ of the filament determines its persistence length $\ell_{\text{p}} = \kappa/k_{\text{B}}T$.

MT bead pairs that are not connected by harmonic springs interact with each other via the repulsive Weeks-Chandler-Andersen (WCA) potential (**Weeks et al., 1971**),

$$U_{\text{wca}} = 4\epsilon \left[ \left(\frac{r_{ij}}{\sigma}\right)^{12} - \left(\frac{r_{ij}}{\sigma}\right)^{6} \right] + \epsilon,$$ (4)

with interaction cutoff $r_{\text{cut}} = 2^{1/6}\sigma$.

## Effective molecular motors

Various theoretical studies have strived to circumvent short time and length scales involved in cytoskeletal dynamics, such as diffusion and active motion of individual motors (**Kruse and Jülicher, 2000**; **Kruse et al., 2005**; **Swaminathan et al., 2010**; **Jia et al., 2008**; **Aranson and Tsimring, 2005**; **Aranson and Tsimring, 2006**; **Salbreux et al., 2009**; **Córdoba et al., 2015**). For example, in the phenomenological flux-force model the motion of MTs in one dimension occurs solely due to the orientation of neighbouring MTs (**Kruse et al., 2004**; **Kruse et al., 2001**). Many two-dimensional models, where MTs are modelled as stiff, polar rods of equal length, take motors into account using a Maxwellian model of inelastic interactions between the rods (**Aranson and Tsimring, 2005**; **Aranson and Tsimring, 2006**; **Jia et al., 2008**; **Swaminathan et al., 2010**). These probabilistic collision rules result in the alignment of rods. Although these models capture the self-organization of MT-motor mixtures into stable patterns of vortices, asters, and smectic bundles, the collision rule approaches do not reproduce the sliding of antialigned MTs described in **Figure 2**.

Sliding of antialigned MTs due to kinesin motors has been identified as key ingredient for cytoplasmic advection in vivo (**Jolly et al., 2010**; **Winding et al., 2016**; **Lu et al., 2016**). Instead of modelling individual motors, in our model MT motion manifests itself as a result of a distribution of motors in an ensemble of orientations between neighbouring MT pairs. Hence, we coarse-grain MT-motor interactions using an effective motor potential that gives a contribution to $U_{\text{mot}}$. A motor bond can form when the crosslinked beads are antialigned, that is the angle that a motor bond vector $\mathbf{m}_{ij}$ makes with the unit orientation vector is acute on both MTs simultaneously, see **Figure 2**,

$$\mathbf{p}_i \cdot \mathbf{m}_{ij} > 0 \quad \text{and} \quad \mathbf{p}_j \cdot \mathbf{m}_{ij} < 0.$$ (5)

Here the orientation vector assigned to bead $i$ on an MT is $\mathbf{p}_i = (\mathbf{r}_{i+1} - \mathbf{r}_i)/|\mathbf{r}_{i+1} - \mathbf{r}_i|$, and the extension of a motor that crosslinks MTs $i$ and $j$ is $\mathbf{m}_{ij}(s_i, s_j) = \mathbf{r}_i(s_i) - \mathbf{r}_j(s_j)$, with the motor heads

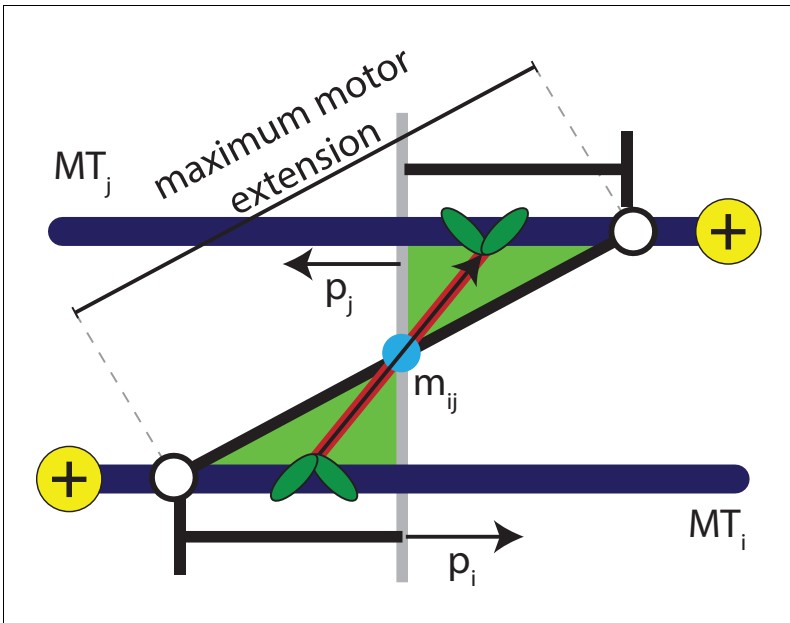

**Figure 2.** Schematic explaining the conditions that satisfy the antialigned motor potential. The vectors, $\mathbf{p}_i$, $\mathbf{p}_j$, and $\mathbf{m}_{ij}$, represent the unit orientation vectors of MT $i$, MT $j$, and the motor vector that crosslinks the beads of adjacent MTs, respectively. The white circles represent the maximum extension of motors between the two MTs.
DOI: https://doi.org/10.7554/eLife.39694.003

bound at the positions $s_i$ and $s_j$ along the contour of the MTs. This is similar to the activity-inducing scenario a dimeric or tetrameric motor (*Ravichandran et al., 2017*) encounters when it crosslinks a pair of antialigned MTs, that is the motor arms are oriented towards the $+$ direction of either cross-linked MT.

Each effective motor is a harmonic spring of equilibrium bond length $d_{\mathrm{eq}} = \sigma$ and stiffness $k_{\mathrm{m}}$ that exists for one simulation time step (Although the force for each motor lasts for a single time step, this duration is not a characteristic time for the motor. Our model describes continuum propulsion forces on MTs imposed by motors. This is supported by *Figure 3*; *Figure 3—figure supplement 1*, which shows that the MT parallel velocity is proportional to the fraction of time that motors are active, independent of the duration of each active phase.). The system is inherently out of equilibrium because the motor bonds occur dependent on the relative orientation of neighboring MTs, and exist and exert forces only for short times, mimicking the ratchet model for molecular motors (*Jülicher et al., 1997*). The potential for a motor with extension $m_{ij} = |\mathbf{m}_{ij}|$ is

$$U_{\mathrm{mot}}(m_{ij}) = \begin{cases} \frac{k_m}{2}(m_{ij} - d_{\mathrm{eq}})^2 & m_{ij} \leq d_{\mathrm{t}} \\ 0 & m_{ij} > d_{\mathrm{t}} \end{cases}, \tag{6}$$

and the motor binding rate is

$$k_{\mathrm{on}}(m_{ij}) = p_{\mathrm{a}} \exp\left(-\frac{U_{\mathrm{mot}}(m_{ij})}{k_{\mathrm{B}} T}\right). \tag{7}$$

Here, $p_{\mathrm{a}}$ controls the probability that an antialigned motor binds (Similarly, we can also implement motors between polar-aligned MTs). Motors bind only for extensions $m_{ij} < d_{\mathrm{t}} = 2\sigma$, when $k_{\mathrm{on}}/p_{\mathrm{a}} > \exp(-1/2)$. This also corresponds well to the experimentally measured length of a kinesin motor (*Kerssemakers et al., 2006*). The motor model described here for two dimensions is analogous to the phenomenological model for one dimension described in *Kruse and Jülicher (2000)*; these one-dimensional calculations show that the relative velocity between two antialigned MTs is a linear function of $p_{\mathrm{a}}$ and $k_{\mathrm{m}}$. Similarly, a kinesin-5 induced effective torque between MTs has been calculated to study forces in the mitotic spindle (*Winters et al., 2018*).

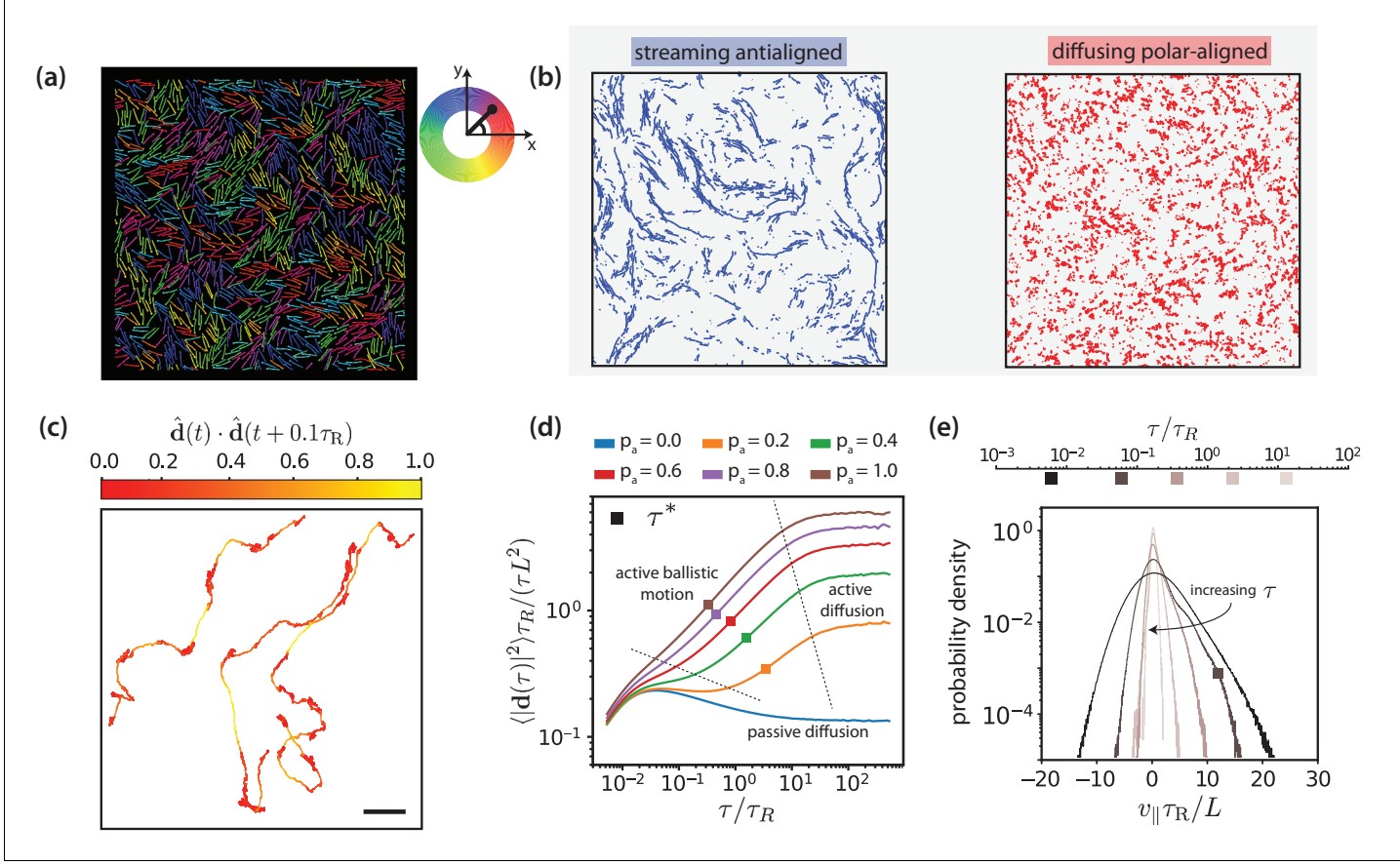

**Figure 3.** Motor-driven and diffusive motion of MTs. (a) Simulation snapshot of MTs organised by effective motors. MTs are coloured based on their orientation according to the colour legend on the right. See corresponding *Video 1*. (b) Trajectories of MTs within a time window of 1.2 $\tau_R$ separated based on the antialigned and polar-aligned categories. See corresponding *Video 2*. (c) Plots of the trajectory of three selected MTs coloured based on the correlation of adjacent steps in their velocity. The entire trajectory is for a time window of 300 $\tau_R$ is the unit vector of MT displacement. The fast-streaming and slow-diffusion modes correspond with the yellow and red parts of the trajectories respectively. The scale bar corresponds to the length of five MTs. See corresponding *Video 3*. (d) MSD/lag time for various levels of activity $p_a$ and MT density $\phi = 0.3$. The time scale of maximal activity, $\tau^*$, calculated from the time of maximal $v_\parallel$ skew is indicated by the squares on the curves. (e) Histogram of parallel velocity for various $\tau$. The curve closest corresponding to the time scale of maximal activity, $\tau^*$, is indicated with a box marker. All figures are for $\phi = 0.3$. (a), (b), (c) and (e) are for $p_a = 1.0$.

DOI: https://doi.org/10.7554/eLife.39694.004

The following source data and figure supplements are available for figure 3:

**Source data 1.** Source data for graphs shown in *Figure 3 (d,e)*.
DOI: https://doi.org/10.7554/eLife.39694.015

**Figure supplement 1.** Parallel velocity $v_\parallel$, extrapolated to $\tau = 0$ as function of the antialigned motor probability $p_a$ for various MT surface fractions $\phi$.
DOI: https://doi.org/10.7554/eLife.39694.005

**Figure supplement 2.** Simulation snapshots at steady state for various antialigned motor probabilities $p_a$ and MT surface fractions $\phi$.
DOI: https://doi.org/10.7554/eLife.39694.006

**Figure supplement 3.** Translational MT mean squared displacements for various antialigned motor probilities $p_a$ and MT surface fractions $\phi$.
DOI: https://doi.org/10.7554/eLife.39694.007

**Figure supplement 4.** Parallel velocity $v_\parallel$ as a function of the time window $\tau$ for various antialigned motor probabilities $p_a$ and MT surface fractions $\phi$.
DOI: https://doi.org/10.7554/eLife.39694.008

**Figure supplement 5.** Maximum parallel MT velocities $v_{\parallel,A}(\tau^*)$ as function of the antialigned motor probability $p_a$ for various MT surface fractions $\phi$.
DOI: https://doi.org/10.7554/eLife.39694.009

**Figure supplement 6.** Histogram of $v_\parallel$ for various MT surface fractions $\phi$ and five time windows $\tau$.
DOI: https://doi.org/10.7554/eLife.39694.010

**Figure supplement 7.** Histogram of $v_\parallel(\tau^*)$ for various $p_a$ and $\phi$.
DOI: https://doi.org/10.7554/eLife.39694.011

**Figure supplement 8.** Probability densities of the MT local polar order parameter $\psi_i$.

*Figure 3 continued on next page*

*Figure 3 continued*
DOI: https://doi.org/10.7554/eLife.39694.012
**Figure supplement 9.** MT parallel velocity distributions.
DOI: https://doi.org/10.7554/eLife.39694.013
**Figure supplement 10.** MT mean squared displacements: computer simulation and experimental data.
DOI: https://doi.org/10.7554/eLife.39694.014

## Results

We characterise several distinct processes comprising the phenomenology of active MT motion that arise because of the sliding of adjacent, antialigned MT pairs.

### Regimes of MT dynamics

*Figure 3* provides an overview of the processes that comprise MT streaming. The fundamental sliding mechanism, which is imposed through the effective motor potential, is the process which occurs at the sliding time $\tau_{N,min}$. The three processes that occur on longer time scales are characterised by the polarity-inversion time $\tau_{Q/2}$, the activity time $\tau^*$, and the collective-migration time $\tau_{N,\max}$. The active-rotation time $\tau_r$ characterises the time when an active MT reaches the end of a polar-ordered domain and changes its orientation.

*Figure 3(a)* and *Video 1* show multiple, small, polar-aligned MT domains with dynamic interfaces of antialigned MTs between them for MT surface fraction $\phi = 0.3$ and motor activity $p_a = 1$. Note that choosing different values for $\phi$ and $p_a$ can have a pronounced effect on the structure of the system (*Figure 3—figure supplement 2*). The domains are formed by polarity sorting (*Wollrab et al., 2018*) and are in dynamic equilibrium due to MTs that perpetually enter and leave them, see *Figure 3(b)* and *Video 2*. Tracing the individual trajectories shows that MT dynamics consists of a fast streaming mode and a slow diffusion mode, see *Figure 3(c)* and *Video 3*. Uncorrelated displacements in time correspond to slow diffusion within a polar-aligned domain of MTs, and correlated displacements correspond to fast, ballistic streaming at interfaces between domains. This leads to a highly dynamic overall MT structure illustrated by *Video 4*. At steady state, the length of antialigned interfaces and the size of polar-aligned domains remain constant. The polarity-inversion time $\tau_{Q/2}$ characterises the duration that MTs stay in polar-aligned bundles or in antialigned streams.

*Figure 3(d)* shows the MT mean squared displacement

$$\mathrm{MSD}(\tau) \equiv \langle|\mathbf{d}_i(t, \tau)|^2\rangle \equiv \langle|\mathbf{r}_i(t + \tau) - \mathbf{r}_i(t)|^2\rangle \quad (8)$$

for MT surface fraction $\phi = 0.3$ and various motor probabilities (We use $m = 1$ and $\gamma = 1$ in simulation units. The single MT center-of-mass

**Video 1.** Steady-state dynamics of the MT-effective motor system shown in *Figure 3(a)* for 100 $\tau_R$. MTs are coloured based on their orientation.
DOI: https://doi.org/10.7554/eLife.39694.016

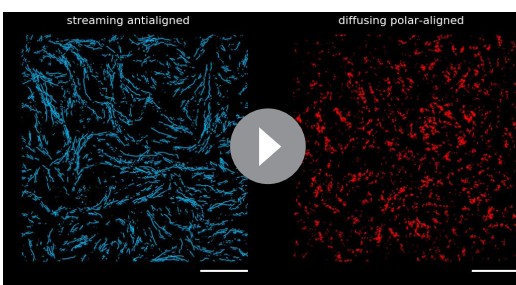

**Video 2.** Streaming motion of antialigned and diffusive motion of polar-aligned MTs for 100 $\tau_R$, corresponding to *Figure 3(b)*. The scale bar corresponds to the length of 10 MTs. Trajectories of MTs are shown within time windows of $1.2\tau_R$.
DOI: https://doi.org/10.7554/eLife.39694.017

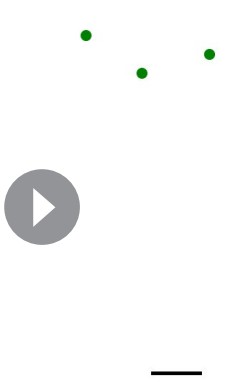

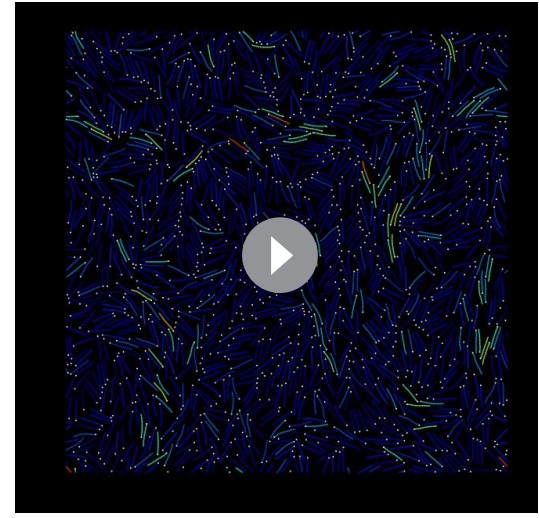

**Video 3.** Center-of-mass trajectories for selected MTs for $300\tau_R$, corresponding to *Figure 3(c)*. Fast streaming and slow diffusion is indicated by yellow and red, respectively. The scale bar corresponds to the length of five MTs.

DOI: https://doi.org/10.7554/eLife.39694.018

**Video 4.** Inhomogeneous dynamics over a period of $100\,\tau_R$. Fast MTs are coloured yellow, and slow MTs are coloured blue.

DOI: https://doi.org/10.7554/eLife.39694.019

dynamics is overdamped for times $t/\tau_R > 2m/\gamma\tau_R \approx 10^{-2}$, which is shorter than the shortest time scale of interest shown in *Figure 3*
*(d)*.). Here, $\mathbf{r}_i(t)$ is the center-of-mass position vector of MT $i$ at time $t$, and $\mathbf{d}_i(t,\tau)$ is the displacement vector of the center-of-mass of MT $i$ between $t$ and $t+\tau$. For passive MTs, $p_a = 0.0$, the ballistic regime $\mathrm{MSD} \propto \tau^2$ at short times due to inertia is followed by a diffusive regime $\mathrm{MSD} \propto \tau$ where the MT velocity is dissipated by the environment. For all simulations at finite $p_a$, we find a superdiffusive regime $10^{-1} \lesssim \tau/\tau_R \lesssim 10^1$ with $\mathrm{MSD} \propto \tau^\alpha$ and $\alpha > 1$ with active ballistic motion. Finally, we find a diffusive regime at long times with an active diffusion coefficient that is much higher than for passive Brownian diffusion. A larger motor activity thus leads to faster filament motion. Filament dynamics is fastest for intermediate MT surface fractions $\phi = 0.3$ and $\phi = 0.4$ (*Figure 3—figure supplement 3*). For smaller densities the required MT-MT contacts are reduced, whereas for large densities ($\phi = 0.6$) excluded volume interactions lead to a larger effective friction hindering filament motion.

*Figure 3(e)* shows the histogram of the MT velocity that is projected on the MT orientation vector $\mathbf{p}(t) = (\mathbf{r}_{n_b}(t) - \mathbf{r}_1(t))/|\mathbf{r}_{n_b}(t) - \mathbf{r}_1(t)|$,

$$v_\parallel(\tau) = \frac{<\mathbf{d}_i(t,\tau) \cdot \mathbf{p}(t)>}{\tau} \qquad (9)$$

for $\phi = 0.3$, $p_a = 1.0$, and various lag times $\tau$, see *Figure 3—figure supplements 1*, *4* and *5* for parallel velocities at other MT surface fractions. At short lag times, the MT displacement is strongly correlated with the MT's initial orientation vector and dominated by thermal noise, giving the largest absolute values for $v_\parallel$. With increasing lag time, the increasing importance of the active motor force is reflected by the increasing asymmetry of the distributions of parallel velocities that are skewed towards positive velocities. At long lag times, the MTs reorient due to active forces, such that both the width of the velocity distribution and the skew again decrease. We characterise the time delay that corresponds to the maximum skew as collective-migration time $\tau_{N,\mathrm{max}}$. This time characterises collective motion of neighbouring MTs with similar orientations that travel in the same direction. The parallel velocity distributions also depend on lag time, filament density, and motor probability (*Figure 3—figure supplements 6*, *7* and *9*).

Finally, the active orientational correlation time $\tau_r$ for MTs is denoted by $\tau_r$. This time characterises the crossover between the active-ballistic and the active-diffusive regime in *Figure 3(d)*. It therefore increases both with increasing size of polar or nematic domains as well as with decreasing rod

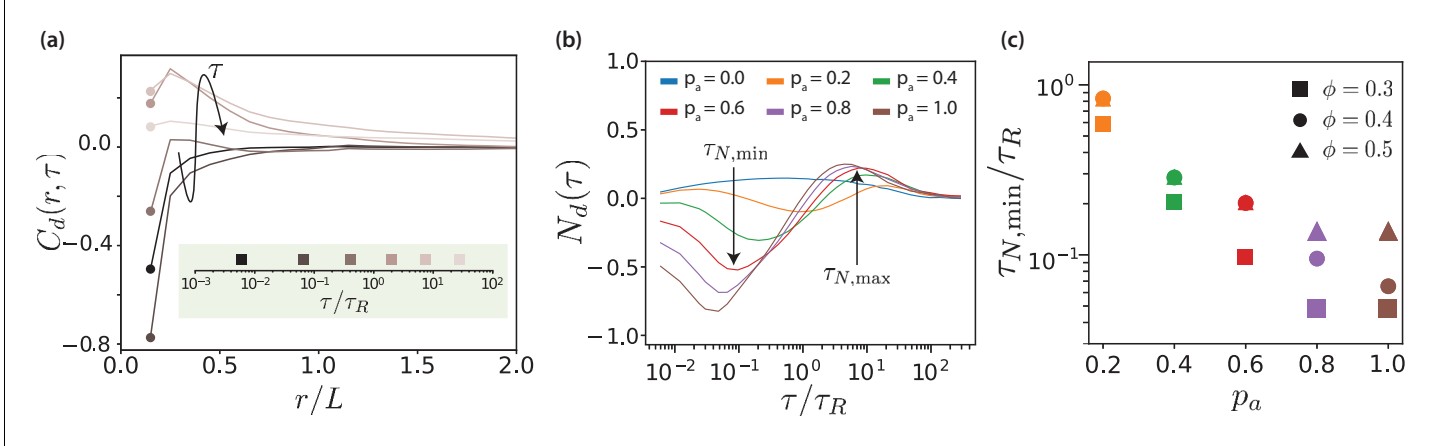

**Figure 4.** Displacement correlations of MTs. (a) Spatio-temporal correlation function $C_{\mathrm{d}}(r, \tau)$ for $\phi = 0.3$ and $p_{\mathrm{a}} = 1.0$, for some selected lag times. The arrow and the colours of the curves indicate increasing lag time. The lag times are picked from a logarithmic scale. (b) Neighbour correlation function $N_{\mathrm{d}}(\tau) = C_{\mathrm{d}}(\sigma, \tau)$ for $\phi = 0.3$ and various $p_{\mathrm{a}}$ values. (c) The sliding time scale indicated by $\tau_{N,\min}$ is shown for various MT surface fractions and $p_{\mathrm{a}}$ values.
DOI: https://doi.org/10.7554/eLife.39694.020

The following source data and figure supplement are available for figure 4:

**Source data 1.** Source data for graphs shown in *Figure 4 (a,b,c)*.
DOI: https://doi.org/10.7554/eLife.39694.022

**Figure supplement 1.** Neighbour displacement correlation function $N_{\mathrm{d}}(\tau)$ for various MT surface fractions $\phi$ and antialigned motor probabilities $p_{\mathrm{a}}$.
DOI: https://doi.org/10.7554/eLife.39694.021

activity at the interfaces, in agreement with the diffusion of tracer particles in *Sanchez et al. (2012)* (*Figure 3—figure supplement 10*).

## Microtubule sliding

The displacement-correlation function,

$$C_{\mathrm{d}}(r, \tau) = \frac{\langle \sum_{i,i \neq j} \mathbf{d}_i \cdot \mathbf{d}_j \delta(r - |\mathbf{r}_i(t) - \mathbf{r}_j(t)|) \rangle_t}{c_0 \langle \sum_{i,i \neq j} \delta(r - |\mathbf{r}_i(t) - \mathbf{r}_j(t)|) \rangle_t}, \tag{10}$$

quantifies both spatial and temporal correlations of MT motion. Here, $\mathbf{d}_i = \mathbf{r}_i(t + \tau) - \mathbf{r}_i(t)$, and $c_0 = \langle \sum_i \mathbf{d}_i^2 / N \rangle_\tau$ is used for normalisation. *Figure 4(a)* shows displacement correlation functions for various lag times. At short times and distances, we find negative displacement correlations due to the effective motor potential, which selectively displaces neighbouring antialigned MTs. These negative correlations decay rapidly in space and do not contribute substantially for $r/L = 1$. At intermediate lag times we find positive displacement correlations with a slower spatial decay, and at long lag times no correlations. In the limit $\tau \to 0$, $C_{\mathrm{d}}$ is the equal-time spatial velocity correlation function (*Wysocki et al., 2014*).

The neighbour displacement correlation function $N_{\mathrm{d}}(\tau) = C_{\mathrm{d}}(\sigma, \tau)$ is defined as the displacement-displacement correlation function at contact $C_{\mathrm{d}}(\sigma, \tau)$ (*Doliwa and Heuer, 2000*; *Wysocki et al., 2014*). *Figure 4(b)* shows neighbour displacement correlation functions for various values of $p_{\mathrm{a}}$ and $\phi = 0.3$. Firstly, for passive systems, $N_{\mathrm{d}}$ is positive for all MT surface fractions but considerably weaker compared to the correlations in active systems. The small positive correlation is due to steric interactions and friction because of to the roughness of MTs (made up of overlapping beads). For active systems, *Figure 4(b)* illustrates that the temporal dependence of $N_{\mathrm{d}}(\tau)$ displays three regimes: for short times, $N_{\mathrm{d}}(\tau)$ is negative and MTs slide antiparallel, for intermediate times, $N_{\mathrm{d}}(\tau)$ is positive and MTs move collectively, and for long times, $N_{\mathrm{d}}(\tau)$ tends to zero and there is no coordinated motion. We focus here on the first regime, whereas the other regimes will be discussed in later sections.

In the short-time regime, $\tau/\tau_R \sim 10^{-1}$, the effective motor potential propels neighbouring antialigned MTs away from each other and $N_{\mathrm{d}}$ is negative. This is aided by higher $p_{\mathrm{a}}$ but hindered by

higher $\phi$, which opposes active motion sterically (*Figure 4—figure supplement 1*). The times $\tau_{N,min}$ at which the minima occur represent maximal MTs propulsion because of effective motor interactions, due to presence of the antialigned neighbours. At this time MTs move a small fraction of their length. In *Figure 4(c)* the sliding times are collected for different MT surface fractions showing that the sliding is strongly enhanced by activity, where $\tau_{N,min}$ decreases approximately exponentially with $p_a$ and increases with surface fraction.

## Polarity inversion of local MT environment

In order to characterise an MT's neighbourhood, we define a pairwise motor partition function (*Gao et al., 2015*; *Ravichandran et al., 2017*),

$$q_{ij} = \rho^2 \sum_{i=1}^{n_b} \sum_{j=1}^{n_b} e^{-U_{mot}(m_{ij})/(k_B T)},$$ (11)

where $\rho = n_b/L$ is the linear density of binding sites on a single MT, and $m_{ij}$ is the extension of the motor bound at positions $s_i$ and $s_j$ on MTs $i$ and $j$, respectively (*Ravichandran et al., 2017*). Local polar order thus weighs pairwise interactions of MTs on the basis of motor binding site availability. It is a function of relative orientation and distance between the beads that are used to model the MTs. Because of the Boltzmann weight, $q_{ij}$ is significant only for pairs of MTs in close proximity. When two MTs are perfectly overlapping each other, $q_{ij} = 1$. When two MTs are sufficiently far away, , $q_{ij} = 0$ because the MTs are outside the motor cut-off range. Since the motor energy $U_{mot}(m_{ij})$ increases quadratically with increasing motor extension, the partition function $q_{ij}$ decays rapidly for increasing distance between the binding sites on the MTs.

The polarity of an MTs environment is quantified by the local polar order parameter $\psi(i)$. MTs within motor cut-off range are defined to be antialigned if $(\mathbf{p}_i \cdot \mathbf{p}_j) < 0$ and polar-aligned if

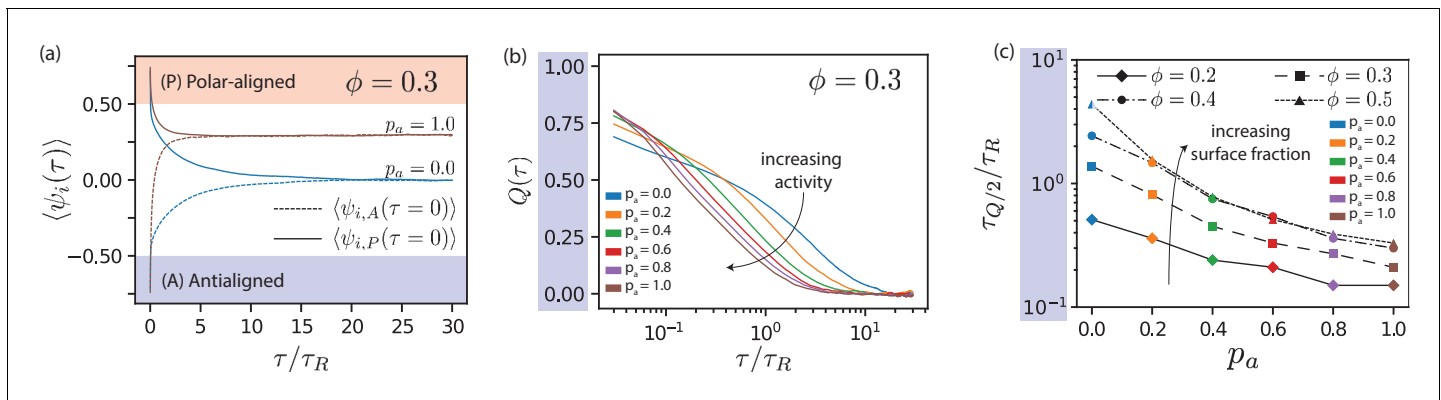

**Figure 5.** Local polar order of MTs. (a) Mean local polar order $\langle \psi_i(\tau) \rangle$ for $p_a = 0.0$ and $p_a = 1.0$ at $\phi = 0.3$, for MTs starting from antialigned (dotted line) and aligned (solid line) environments at $\tau = 0$. (b) Deviation of local polar order $Q(\tau)$ for $\phi = 0.3$ for various $p_a$ for antialigned MTs. (c) Relaxation time for the polar order parameter, $\tau_{Q/2}$ for various $p_a$ and $\phi$, estimated by the time for $Q$ to decrease to half its initial value.

DOI: https://doi.org/10.7554/eLife.39694.023

The following source data and figure supplements are available for figure 5:

**Source data 1.** Source data for graphs shown in *Figure 5 (a,b,c)*.
DOI: https://doi.org/10.7554/eLife.39694.027

**Figure supplement 1.** MTs coloured based on their local polar order parameter $\psi_i$ for $p_a = 1.0$, $\phi = 1.0$.
DOI: https://doi.org/10.7554/eLife.39694.024

**Figure supplement 2.** Deviation from local polar order $Q(\tau)$ as function of the lag time $\tau$ for various antialigned motor probabilities $p_a$ for polar-aligned MTs.
DOI: https://doi.org/10.7554/eLife.39694.025

**Figure supplement 3.** Mean local polar order parameter of MTs at long times, $\langle \psi_\infty \rangle$, for various surface fractions $\phi$ and antialigned motor probabilities $p_a$.
DOI: https://doi.org/10.7554/eLife.39694.026

$(\mathbf{p}_i \cdot \mathbf{p}_j) \geq 0$. By taking the sum of all interacting MTs $j \neq i$ with MT $i$ (**Gao et al., 2015**; **Ravichandran et al., 2017**), we ensure that the local polar order parameter

$$\psi(i) = \frac{\sum_{j \neq i}(\mathbf{p}_i \cdot \mathbf{p}_j)q_{ij}}{\sum_{j \neq i}q_{ij}} \tag{12}$$

depends on the polarity of the neighbourhood of MT $i$. Here $q_{ij}$ is given by **Equation 11**. The environment of the MT can now be classified into polar (subscript-$P$, $0.5 < \psi(\tau) < 1$), antipolar (subscript-$A$, $-1 < \psi(\tau) < -0.5$), and mixed (subscript-$M$, $-0.5 < \psi(\tau) < 0.5$), see **Figure 3—figure supplement 8** and **Figure 5—figure supplement 1**. When a single MT's environment changes from predominantly antipolar to polar its active motion is stopped and it only moves diffusively.

By tracking changes in $\psi_i$ for single MTs, see **Video 2**, we measure the time that MTs spend in antialigned or polar-aligned environments for various values of $p_a$ and $\phi$. The change in local polar order of MT $i$ can be written as

$$\Delta\psi_i(\tau) = \psi_{i,0} - \psi_i(\tau), \tag{13}$$

where $\psi_{i,0} = \psi_i(\tau = 0)$. **Figure 5(a)** shows $\langle\psi_{i,A}(\tau)\rangle$ and $\langle\psi_{i,P}(\tau)\rangle$ for $p_a = 1$ and $p_a = 0$. In both cases, we find that $\langle\psi_{i,A}(\tau)\rangle$ increases with time, indicating antialigned MTs leaving their antialigned environments, and that $\langle\psi_{i,P}(\tau)\rangle$ decreases with time, indicating polar-aligned MTs leaving their polar-aligned environments. At long times, $\psi_{i,A}$ and $\psi_{i,P}$ converge to the long-time mean $\langle\psi_{i,\infty}\rangle = 0$ for passive systems, and to $\langle\psi_{i,\infty}(\tau)\rangle > 0$ for active systems. The time scale for relaxing $\langle\psi_i\rangle$ to the equilibrium value is, as expected, shorter for the active than for the passive system.

In order to quantify the change in $\langle\psi_i\rangle$, we construct the deviation of local polar order,

$$Q(\tau) = 1 - \frac{\langle\Delta\psi_i(\tau)\rangle}{\langle\psi_{i,0}\rangle - \langle\psi_{i,\infty}\rangle}. \tag{14}$$

$Q(\tau)$ for the antialigned MTs is shown in **Figure 5(b)**; the lag time for that $Q(\tau)$ reaches half its initial value is $\tau_{Q/2,A}$. While MTs that stay within a polar-ordered domain determine the offset for $\phi_{i,A}$ at long times, only MTs entering polar aligned domains determine $\tau_{Q/2,A}$. **Figure 5(c)** shows that $\tau_{Q/2,A}$ decreases almost exponentially with $p_a$ and increases with $\phi$. In stationary state, the time scales for the inversion of local polar order of initially antialigned and initially polar-aligned MTs are equal, $\tau_{Q/2,A} = \tau_{Q/2,P}$, compare **Figure 5(b)** and **Figure 5—figure supplement 2**. The dependence of $\langle\psi_{i,\infty}\rangle$ $\phi$ and $p_a$ is shown in **Figure 5—figure supplement 3**.

## Maximal activity

The mean squared displacements of MTs are ballistic, diffusive, or superdiffusive depending on the lag time, see **Figure 3(d)**. This is reflected in the distributions of the parallel velocity $v_\parallel$, see **Equation 9** and **Figure 3(e)**. The $v_\parallel$ distributions become increasingly asymmetric with increasing lag time when active propulsion dominates over Brownian motion for antialigned MTs–and again less asymmetric when the lag time is further increased and orientational memory is lost. Because of the high number of parallel MTs in our simulations the position of the main peak is at $\langle v_\parallel\rangle = 0$ as expected for passive MTs. A skew of the distribution can then be understood as a superposition of a high peak of non-propelled polar-aligned MTs and a small peak that is shifted to positive values of $v_\parallel$ for antialigned MTs, see also **Videos 2** and **4**.

In **Figure 6**, we plot skews of $v_\parallel$ distributions (**Figure 3(e)**) as function of lag time for various $p_a$ values for $\phi = 0.3$. Both maximal skew and maximal lag time for that we detect finite skews increase with increasing $\phi$, **Figure 6—figure supplement 1**. The lag time at which the skew of the $v_\parallel$ distribution is maximal is defined as the activity time $\tau^*$, where the ratio of the displacements due to active forces to the thermal displacements is largest. This activity time $\tau^*$ falls into the regime where the MSD is most superdiffusive, see **Figure 3(d)**. As shown in **Figure 6(b)**, $\tau^*$ exponentially decreases with increasing $p_a$ and increases with $\phi$. Increasing motor concentration, and thus a higher amount of active forces in the system, is akin to exponentially shifting the activity time to shorter values.

The proportion of aligned (passive) and antialigned (propelled) MTs depends strongly on the area fraction, where the number of antialigned MTs decreases with surface fraction $\phi$. This corresponds

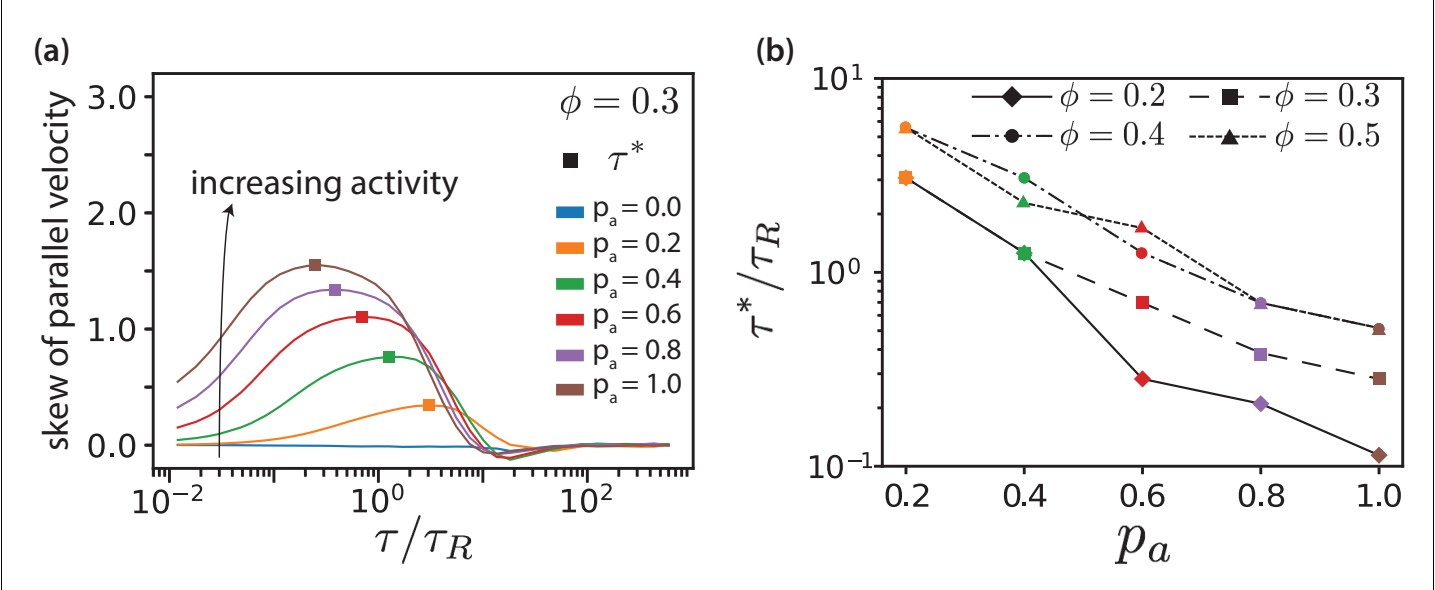

**Figure 6.** MT parallel velocity distributions. (a) Skew of parallel velocity ($v_\parallel$) distribution computed as function of lag times for different $p_a$ for $\phi = 0.3$. The probability distributions that correspond to the maximal skew are shown in *Figure 3—figure supplement 6* together with distributions for few other lag times. (b) Lag time at which maximal skew is observed in the $v_\parallel(\tau)$ distribution (compare *Figure 1*). The ordinate is log-scaled to show that $\tau^*$ is exponentially decreasing with $p_a$.

DOI: https://doi.org/10.7554/eLife.39694.028

The following source data and figure supplements are available for figure 6:

**Source data 1.** Source data for graphs shown in *Figure 6 (a,b)*.

DOI: https://doi.org/10.7554/eLife.39694.032

**Figure supplement 1.** Skews $\alpha_3$ of parallel velocity distributions ($v_\parallel$) (Fig.

DOI: https://doi.org/10.7554/eLife.39694.029

**Figure supplement 2.** Ratios of MT populations in environments with different local polar order.

DOI: https://doi.org/10.7554/eLife.39694.030

**Figure supplement 3.** First three moments of the parallel velocity ($v_\parallel$) distribution.

DOI: https://doi.org/10.7554/eLife.39694.031

to larger domain sizes and less interfaces with increasing $\phi$, see *Figure 3—figure supplement 2* and *Figure 6—figure supplement 2*. Further, we notice that for all surface fractions $\phi$ increasing $p_a$ widens the $v_\parallel(\tau)$ distributions, see *Figure 3—figure supplement 6*. The widening of the distribution becomes less pronounced with increasing $\phi$. The shifting of the negative part of the $v_\parallel$ distribution ($v_\parallel(\tau^*) < 0$) to more negative values with increasing $p_a$ is because $\tau^*$ decreases simultaneously.

## Collective migration

The observables discussed so far characterise the motion of individual MTs. They only take collective effects into consideration indirectly, for example via the asymmetry of the $v_\parallel$ distribution for polar-aligned MTs. For a direct discussion of the time scale of collective effects, we return to *Figure 4(b)*. For times around $\tau/\tau_R = 10$, in the intermediate time regime, we observe a positive neighbour displacement correlation. This behaviour is altogether absent at low surface fractions, $\phi = 0.2$, but for larger $\phi$ the positive correlations increase with increasing $\phi$. This suggests that neighbouring MTs in a particular stream (likely polar-aligned) travel in the same direction. These polar-aligned MTs will collectively migrate in the same direction because they are in a similarly antialigned environment, that is at the same interface with another domain. Correlations in their motion can only manifest at longer lag times, since at short lag times the correlation contribution will be dominated by fast-moving antialigned MTs. We denote the lag time for the maximum of $N_d(\tau)$, when collective migration occurs, $\tau_{N,max}$.

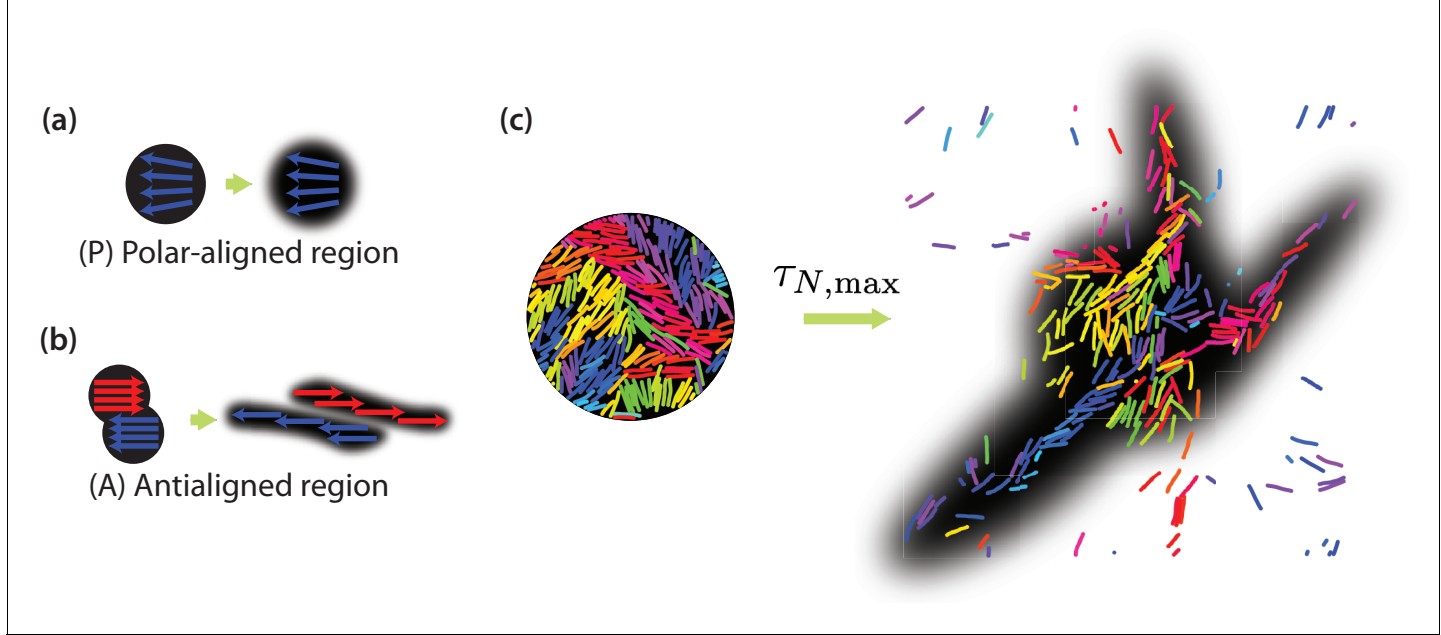

**Figure 7.** Collective motion of MTs. Schematic of expected evolution of photobleached regions in (**a**) polar-aligned and (**b**) antialigned regions. (**c**) Selectively visualised MTs in a circular region within the simulation box, and their evolution after a time of $\tau_{N,\mathrm{max}}$, for $\phi = 0.4$ and $p_a = 1.0$. The black backgrounds are predictions of FRAP results.

DOI: https://doi.org/10.7554/eLife.39694.033

The following figure supplement is available for figure 7:

**Figure supplement 1.** FRAP-like predictions for various MT surface fractions $\phi$.

DOI: https://doi.org/10.7554/eLife.39694.034

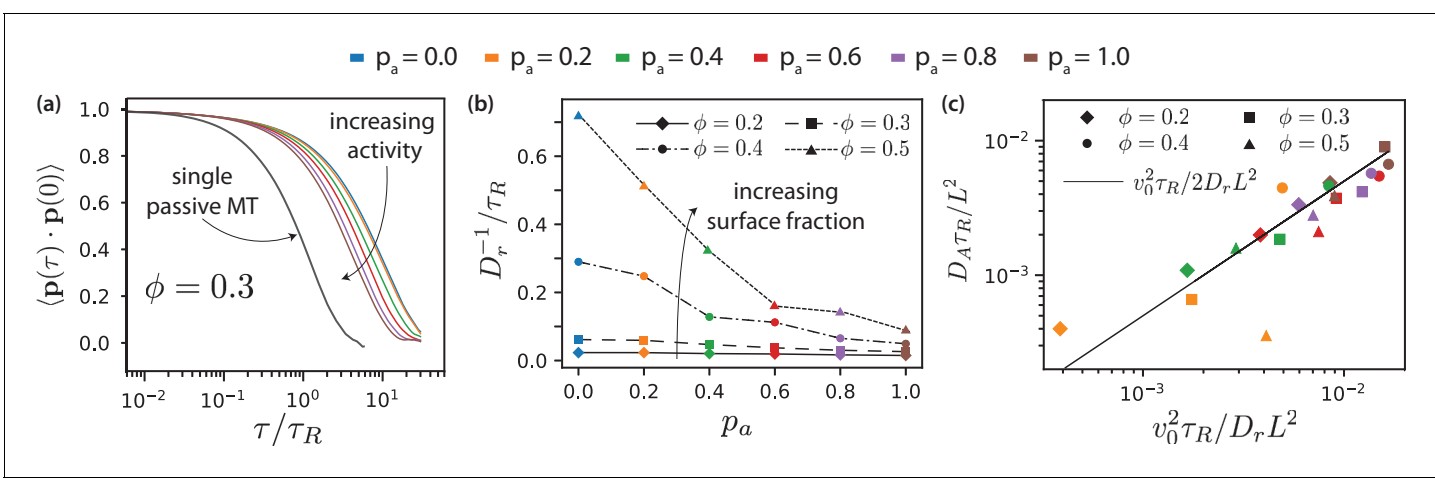

**Figure 8.** MT orientational correlation and active diffusion. (**a**) Orientational correlation function for $\phi = 0.3$ for various antialigned motor probabilities $p_a$. (**b**) Inverse of rotational diffusion, $\tau_r$ for various antialigned motor probabilities $p_a$ and surface fractions $\phi$ (**c**) Active diffusion coefficient $D_A$ for $p_a = 1$.

DOI: https://doi.org/10.7554/eLife.39694.035

The following source data is available for figure 8:

**Source data 1.** Source data for graphs shown in **Figure 8** (a,a-one filament,b,c).

DOI: https://doi.org/10.7554/eLife.39694.036

In order to explicitly show that positive neighbour displacement correlations observed in the intermediate time regime are due to collective migration of similarly oriented MTs, we can predict the results of photobleaching or photoactivation experiments (*Gao et al., 2015*; *Mitchison, 1989*; *Hush et al., 1994*). Experimentally, in a photobleaching experiment a high-intensity laser beam can be used to inactivate fluorescent molecules in a circular region (*Axelrod et al., 1976*). The time evolution of the distribution of the light-inactivated regions gives clues about the underlying mechanisms which mediate this motion. *Figure 7(a)* illustrates that we expect little or no MT sliding to occur in a polar-aligned region, and the photobleached area maintains its shape. In *Figure 7(b)*, however, the photobleached area is antialigned and we expect bundles of antialigned domains to slide away, causing the photobleached spot to separate into two elongated regions. In our simulations, we perform a similar measurement, where instead of inactivating regions to inhibit fluorescence, we selectively label MT beads within a certain region. We then track their locations for $t = \tau_{N,max}$ and investigate their displacements. MTs move in response to the effective motor potential and form streams. In *Figure 7(c)*, we visualise a four-MT length radius circular area for $\phi = 0.4$ and $p_a = 1.0$. The structures become more diffuse for $\phi = 0.3$ and more compact for $\phi = 0.5$ (*Figure 7— figure supplement 1*).

## Active rotation

The longest relevant time scale for the MT dynamics is that of active rotational motion characterised by the orientational correlation function

$$\langle \mathbf{p}(t) \cdot \mathbf{p}(t+\tau) \rangle = e^{-\tau/\tau_r}. \tag{15}$$

By fitting *Equation 15* to the simulation data (*Figure 8(a)*), we obtain the transition time to long-time active diffusive behaviour, $\tau_r$. *Figure 8(b)* shows $\tau_r$ for various $p_a$. For passive systems ($p_a = 0$), $\tau_r$ increases with MT surface density $\phi$. For active systems, $\tau_r$ decreases with increasing $p_a$ and with decreasing $\phi$. In the nematic state, MTs are no longer able to rotate freely as in the isotropic case. The decrease of $\tau_r$ with increasing $p_a$ is more pronounced at higher MT surface fractions. Smaller values of $\tau_r$ correspond to smaller domain sizes. In larger domains, the streams appear at interfaces

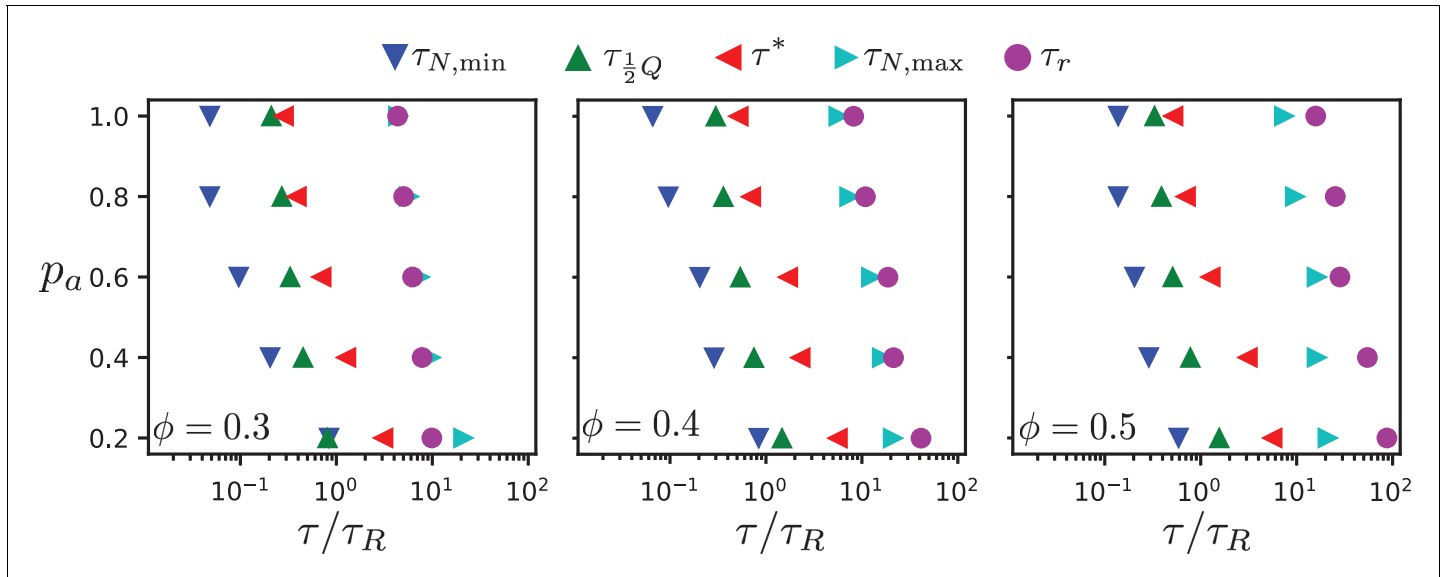

**Figure 9.** Chronology of MT streaming. Events from antialigned MT propulsion to MT rotation (left to right) which make up the streaming process, for various antialigned motor probabilities $p_a$ and surface fractions $\phi = 0.3$, $\phi = 0.4$, and (c) $\phi = 0.5$ as indicated.

DOI: https://doi.org/10.7554/eLife.39694.037

The following source data is available for figure 9:

**Source data 1.** Source data for graphs shown in *Figure 9 (a,b,c)*.

DOI: https://doi.org/10.7554/eLife.39694.038

**Table 1.** Table of time scales involved in MT dynamics.

The time scales reported are approximate values for various antialigned motor probabilities $p_a$ and surface fractions $\phi$.

|  | Symbols | Time scale ($\tau_R$) | Derivation |
|---|---|---|---|
| Passive diffusion | $\tau_D$ | $\tau < 10^{-1}$ | Slope of MSD $\approx 1$ |
| Antialigned propulsion | $\tau_{N,\min}$ | $\tau \approx 10^{-1}$ | Minimum of $N_d$ |
| Streaming | $\tau_{Q/2}$ | $10^{-1} < \tau < 10^0$ | $\langle \psi_i \rangle$ decay time |
| Maximal skew | $\tau^*$ | $10^0 < \tau < 10^1$ | Maximum skew of $\mathbf{p}_0 \cdot \mathbf{d}$ |
| Collective migration | $\tau_{N,\max}$ | $\tau \approx 10^1$ | Maximum of $N_d$ |
| Active rotation | $\tau_r$ | $\tau > 10^1$ | Orientational correlation time |

DOI: https://doi.org/10.7554/eLife.39694.039

between polar-ordered domains and antialigned MTs. The streams extend in the same direction over larger lengths, for longer times, and MTs do not rotate away from their initial orientation as quickly. Also, MTs that are trapped in aligned MT bundles are less likely to exit their environments and their rotational diffusion is smaller for higher $\phi$ and lower $p_a$. Only for $\tau > \tau_r$ the MTs show again diffusive motion with an active diffusion coefficient $D_A \propto v_\parallel^2 \tau_r$, see **Figure 3(d)**.

## Discussion

In our two-dimensional simulation model, dipolar effective motor forces that drive antialigned MT pairs are sufficient to bring about MT streams which are perpetually created and annihilated, akin to MT streaming in biology. Processes that occur on several characteristic times characterise streaming in our MT-motor mixtures: the characteristic time $\tau_{N,min}$ corresponds to the strongest anti-aligned motion of neighbouring MTs, the time $\tau_{Q/2}$ that an MT stays within a stream, the time $\tau^*$ that corresponds to maximal skew of the MT velocity distribution, the collective migration time $\tau_{N,max}$ that characterises maximal directed active motion, and the active rotation time $\tau_r$ that corresponds to single rods traveling the distance of a polar-aligned domain when they loose their orientational memory. **Figure 9** and **Table 1** summarise our findings. (We do not show $\phi = 0.2$, because there is no evidence of streaming in these systems, and the chronology of events is not consistent with those observed for higher surface fractions.)

All characteristic times increase with decreasing motor attachment probability $p_a$. We expect the sliding time $\tau_{N,min}$ and the active rotation time $\tau_r$ to diverge with vanishing motor attachment probability, while the collective migration time $\tau_{N,max}$ and the polarity-inversion time $\tau_{Q/2}$ attain finite values due to thermal motion and steric interaction between the MTs. The activity time $\tau^*$ is not defined for passive systems. Overall, the characteristic times increase with increasing MT surface density, because the lifetime and the coherence of the streams increases. However, our simulations also reveal details of the multi-scale process of streaming in MT-motor mixtures. For example, the time for an MT to transition from a polar-aligned to an antialigned environment is similar to the sliding time for low motor forces, where thermal motion dominates, and to the activity time for high motor forces when the streams are more stable.

The closest 'bottom-up' experimental system to our simulation model is the in vitro model system of microtubule bundles, kinesin complexes, and depletants at the oil-water interface, investigated in *Sanchez et al. (2012)*. A detailed quantitative comparison is currently not possible because the characteristic length scales in simulations and experiment are quite different. The depletion-induced MT bundle formation in the experiments leads to a characteristic length scale of the order of $10\mu m$, whereas the MTs in the simulations have lengths below $1\mu m$. However, on a more qualitative level interesting correspondences are revealed. The transition between diffusive and ballistic MSDs in the simulations has also been reported for the in vitro model system (*Sanchez et al., 2012*). This allows

**Table 2.** Parameter values used in the simulations.

| Parameter | Symbol | Value | Notes/Biological Values |
|---|---|---|---|
| Thermal energy | $k_{\mathrm{B}}T$ | $4.11\,\mathrm{pN\,nm}$ | room temperature |
| MT length | $L$ | $0.625\,\mu m$ | $2.5 \pm 1.4\,\mu m$ (*Howard et al., 1989*) |
| MT diameter | $\sigma$ | $25\,\mathrm{nm}$ | (*Chrétien and Wade, 1991*) |
| MT bond angle constant | $\kappa$ | $2.055 \times 10^4\,\mathrm{pN\,nm^2}$ | rigid MTs |
| MT bond spring constant | $k_{\mathrm{s}}$ | $13.15\,\mathrm{pN/nm}$ | preserves MT length (*Isele-Holder et al., 2015*) |
| Dynamic viscosity | $\eta$ | $1\,\mathrm{Pa\,s}$ | viscosity of cytoplasm (*Wirtz, 2009*) |
| Characteristic energy of WCA potential | $\epsilon$ | $4.11\,\mathrm{pN\,nm}$ | (*Bates and Frenkel, 2000*; *Bolhuis and Frenkel, 1997*; *McGrother et al., 1996*) |
| Motor spring constant | $k_{\mathrm{m}}$ | $6.6 \times 10^{-3}\,\mathrm{pN/nm}$ | $0.33\,\mathrm{pN/nm}$ per kinesin (*Coppin et al., 1995*), high number of effective motors |
| Equilibrium motor length | $d_{\mathrm{eq}}$ | $25\,\mathrm{nm}$ | MT-MT distance at contact |
| Motor dwell time | $\delta t$ | $4.16 \times 10^{-4}\,\mathrm{s}$ | |

DOI: https://doi.org/10.7554/eLife.39694.040

the comparison of the active diffusive regime for times $\tau > \tau_r$ and for lengths longer than a typical domain size. Whereas the motion in suspensions of passive MTs is diffusive, a ballistic regime at large lag times is found for increasing concentrations of active motors (simulations) and for increasing ATP concentration (experiment).

Some of our results can be used to interpret experimental results in vivo. For example, using Particle Image Velocimetry (PIV) in *Drosophila* cells, fluid velocity distributions have been measured for wild-type oocytes and those lacking *pat1*, a protein required for kinesin heavy chain to maximise its motility (*Ganguly et al., 2012*). The main peak is close to a velocity of 10 nm/s, which hints that the majority of the MTs are propelled. As in our simulations, heavy tails in the velocity distribution have been reported in the experiments. A comparison of the experimental data for wild-type and *pat1*-deficient systems showed that for the wild-type system the mean speed was slower and the velocity distribution had heavier tails. This qualitatively agrees with our findings for varying $p_{\mathrm{a}}$. It was suggested in *Ganguly et al. (2012)* that the heavy tails in the velocity distribution of the cytosol reflect a combination of an underlying distribution of motor speeds, and a complex MT network geometry. From our simulations, we conclude that neither a complex three-dimensional cytoskeletal geometry nor a combination of different motor speeds are required to reproduce cytoskeletal velocity distributions with heavy tails.

We have studied the characteristic times of MT-motor dynamics relevant for the cytoskeleton using a coarse-grained motor model and Langevin Dynamics simulations in the overdamped regime. This allows us to access both the single-MT level as well as the collective-MT level. In previous studies that use a similar coarse-graining technique for the motor activity, the focus has been on understanding and capturing biologically relevant cytoskeletal structures (*Aranson and Tsimring, 2005*; *Jia et al., 2008*). Here, for the first time, we have decomposed the time scales of activity from single MTs to system-scale ordering and streaming.

MT advection has also been analysed using photoconversion in interphase *Drosophila* S2 cells, where MTs were observed to buckle and loop (*Jolly et al., 2010*). MT motion was visualised by photoconverting a circular region within the cell. These MTs were observed over a 7 minute period, during which 36% of the MTs were determined to be motile. It was observed that MTs spent most of the time not moving, but underwent abrupt long-distance streaming. They were found to achieve velocities up to 13 $\mu$m/min, during these bursts of active motion. These observations are very similar to those in our simulations, where MTs spend most of their times in stable polar-aligned bundles, but when in contact with an antialigned MTs coherently stream over large distances. We find similar fractions of motile MTs between 30% and 40% also for $\phi = 0.3$ in our simulations. Our study provides the basis for a more detailed quantitative comparison with experiments because the model can be easily extended to include further relevant aspects, such as a 3D cytoskeletal network, cross-linking proteins, and cellular confinement.

**Table 3.** Dimensionless parameters and ranges of the values used in the simulations.

| Parameter | Symbol | Value |
|---|---|---|
| MT surface fraction | $\phi$ | $0.2 - 0.5$ |
| MT aspect ratio | $L/\sigma$ | 10 |
| Reduced MT bond angle stiffness | $\kappa\sigma/k_B T$ | 200 |
| Reduced MT persistence length | $\ell_\mathrm{p}/L$ | 200 |
| MT bond spring constant | $k_\mathrm{s}\sigma^2/k_\mathrm{B}T$ | 2000 |
| Reduced motor spring constant | $k_\mathrm{m}\sigma^2/k_\mathrm{B}T$ | 1 |
| Reduced motor equilibrium length | $d_\mathrm{eq}/\sigma$ | 1 |
| Antialigned motor probability | $p_\mathrm{a}$ | $0 - 1.0$ |
| Reduced single-bead friction | $\gamma/(k_\mathrm{m}\delta t)$ | 171.6 |
| Reduced system size | $L_\mathrm{b}/L$ | $16 - 25$ |

DOI: https://doi.org/10.7554/eLife.39694.041

Our simulations show collective migration of MTs that is maximal at $\tau_{N,\max}$. Using a FRAP-like visualisation of our data, we find elongated MT stream patterns similar to those observed in experiments (*Jolly et al., 2010*). This confirms that similarly oriented MTs move colletively in the same stream. Based on the polarity-sorting mechanism of MTs, qualitatively similar FRAP results have previously been predicted using computer simulations (*Gao et al., 2015*). Experimental studies of a system on various length and time scales should allow testing the chronology that we predict. For example, systems with different fractions of fluorescent MTs with fixed lengths should give access to both collective as well as single-MT dynamics, for example using FRAP/photoactivation for systems with many labeled MTs to quantify collective dynamics and confocal microscopy for systems with few labeled MTs to investigate correlations in single-MT motion.

We have studied collective motion in active gels based on single MTs. Our spatio-temporal displacement correlation functions show that antialigned MTs slide away from each other in opposite directions for short time windows, while in agreement with experiments positive correlations occur for long time windows (*Ganguly et al., 2012*). Our two-dimensional simulations resemble systems close to an interface that have been used to experimentally study hierarchically assembled active matter (*Sanchez et al., 2012*). They also lay the foundations for future studies of 3D systems and have allowed us to test parameter regimes using less computationally expensive, two-dimensional systems. Furthermore, although we observe streaming without hydrodynamics, hydrodynamic interactions may still be an important player for motor-MT systems, which can be investigated in future studies.

To summarize, our results provide a direct handle to fully characterise MT streaming over a wide range of time and length scales. Future experimental studies using modern microscopy techniques may allow testing our predictions. Future theoretical and computer simulation studies may provide further insights, such as the importance of the aspect ratio of the MTs, the presence of motors between polar-aligned MTs, and the effect of crosslinkers important for buckling and looping of flexible MTs.

## Materials and methods

### Langevin dynamics

We simulate the MT-motor systems in two dimensions using periodic boundary conditions. The motion of the beads is described by the Langevin equation,

$$m\frac{d^2\mathbf{r}_i}{dt^2} = -\nabla U_i + F_\mathrm{mot} - \gamma\frac{d\mathbf{r}_i}{dt} + \xi_i(t), \qquad (16)$$

where $\mathbf{r}_i$ is the position of bead $i$, $m$ is the mass of a bead, $\gamma$ is the friction coefficient of the solvent for bead motion, $F_\mathrm{mot} = -\nabla U_\mathrm{mot}$ is the active motor force and $\xi_i$ is the Gaussian-distributed thermal force. The friction coefficient can be estimated using the Stokes friction $\gamma = 6\pi\eta R$ for a spherical

particle with radius $R$ in a solvent with viscosity $\eta$. The thermal forces $\xi_i$ have $\langle \xi_i \rangle = 0$ and, from the fluctuation-dissipation theorem,

$$\langle \xi_\alpha(t)\xi_\beta(t') \rangle = 2\gamma k_B T \delta_{\alpha\beta} \delta(t - t'), \tag{17}$$

where $k_B$ is the Boltzmann constant, T is the temperature, and $\xi_\alpha(t)$ is the $\alpha$-th component of the vector $\xi_i(t)$.

Langevin dynamics simulations allow the use of larger time steps compared with Brownian dynamics simulations without a particle mass. The friction constant $\gamma$ and bead mass $m$ are chosen such that the center-of-mass motion of passive MTs at the same density is diffusive at length scales larger than a fraction of the MT length and at time scales $\tau/\tau_R \geq 0.01$ (see SI), such that passive MTs only move ballistically at times shorter than the relevant times.

The simulation package LAMMPS has been employed to perform the simulations (*Plimpton, 1995*), see *Source code file 1*.

## System parameters

Each simulation consists of $n_\mathrm{f} = 1250$ semiflexible filaments with aspect ratio 10, each made up of $n_\mathrm{b} = 21$ overlapping beads, which reduces the friction of the otherwise corrugated MTs (*Abkenar et al., 2013*; *Isele-Holder et al., 2015*). The MT surface fraction $\phi = n_\mathrm{f} L\sigma/L_\mathrm{b}^2$ is controlled by adjusting the box size $L_\mathrm{b}$. Our effective-motor model is a coarse-grained model and individual effective motors in the simulations may not represent individual motors in experiments. However, the system parameters are based on those of biological systems, see *Table 2*.

The WCA potential is used with the interaction cutoff at $2^{1/6}\sigma$, such that the potential between MTs is purely repulsive. The bond stiffness is large, such that the contour length of the MTs remains approximately constant throughout a simulation run. The angle potential is chosen such that MTs are rigid; the persistence length is $\ell_p = 200L$. We use a time step of duration $\delta t = 5.31 \times 10^{-6}\tau_R$. Each run for a particular parameter set consists of $3\,10^7$ time steps.

We nondimensionalise the key parameters using the MT diameter $\sigma$ or length $L$, thermal energy $k_\mathrm{B}T$, and the single-MT rotational diffusion time $\tau_R$, see *Table 3*.

## Acknowledgements

OD and GS acknowledge support by the International Helmholtz Research School of Biophysics and Soft Matter (IHRS BioSoft). CPU time allowance from the Jülich Supercomputing Centre (JSC) is gratefully acknowledged.

## Additional information

### Funding

| Funder | Grant reference number | Author |
| --- | --- | --- |
| International Helmholtz Research School of Biophysics and Soft Matter | Graduate Student Fellowship | Özer Duman Guglielmo Saggiorato |

The authors declare that there was no funding for this work.

### Author contributions

Arvind Ravichandran, Investigation, Visualization, Methodology, Writing—original draft, Writing—review and editing, Code development-simulation, Code development-data analysis, Code development-visualization; Özer Duman, Investigation, Methodology, Code development-data analysis, Code development-visualization; Masoud Hoore, Methodology, Writing—original draft, Code development-simulation; Guglielmo Saggiorato, Conceptualization, Methodology; Gerard A Vliegenthart, Thorsten Auth, Conceptualization, Investigation, Methodology, Writing—original draft, Writing—review and editing; Gerhard Gompper, Conceptualization, Resources, Investigation, Methodology, Writing—review and editing

## Author ORCIDs

Arvind Ravichandran  http://orcid.org/0000-0001-6247-5229
Masoud Hoore  http://orcid.org/0000-0003-1442-4739
Gerard A Vliegenthart  https://orcid.org/0000-0003-2459-8652
Thorsten Auth  http://orcid.org/0000-0002-6618-2316
Gerhard Gompper  http://orcid.org/0000-0002-8904-0986

## Decision letter and Author response

Decision letter https://doi.org/10.7554/eLife.39694.046
Author response https://doi.org/10.7554/eLife.39694.047

## Additional files

### Supplementary files

• Source code file 1. Source code that adds effective motors to simulations of semiflexible filaments using the open-source Large-scale Atomic/Molecular Massively Parallel Simulator (LAMMPS).
DOI: https://doi.org/10.7554/eLife.39694.042

• Transparent reporting form
DOI: https://doi.org/10.7554/eLife.39694.043

### Data availability

Source code and input files required to simulate microtubule-effective motor systems using the freely available, open-source Large-scale Atomic/Molecular Massively Parallel Simulator (LAMMPS) have been provided.

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
