## [Decision Letter]

Thank you for sending your article entitled "Chronology of motor-mediated microtubule streaming" for peer review at *eLife*. Your article is being evaluated by three peer reviewers, one of whom is a member of our Board of Reviewing Editors, and the evaluation is being overseen by Anna Akhmanova as the Senior Editor.

Given the list of essential revisions, including new experiments, the editors and reviewers invite you to respond within the next two weeks with an action plan and timetable for the completion of the additional work. We plan to share your responses with the reviewers and then issue a binding recommendation.

Summary:

This article describes a computational study of a model for cytoplasmic streaming, motivated by the phenomenology found in *Drosophila* oocytes. The authors introduce a coarse-grained model in which details of the molecular motors' activity that can lead to sliding between adjacent microtubules are subsumed into an effective orientation-dependent potential. The microtubules are modelled as linked spheres, and the whole setup is described by a Langevin equation. The results include many aspects of the correlation functions between the microtubules, with different regimes characterized by different types of alignments. Probability distribution functions of speeds are computed, of the kind that can be measured experimentally. The authors make some contact with experimental studies of streaming using PIV and advance the hypothesis that heavy-tailed velocity distributions arise without the previously conjectured need for varying motor speeds or complex cytoskeletal geometry.

Essential revisions:

1) Whereas the mechanism postulated by authors does produces streaming in the in silico system studied here, it is not guaranteed that the same mechanism occurs inside the cell or in vitro. Streaming occurs in a vast range of systems, from plants upwards, and in many of these systems the filaments are both organized and unchanging in their conformations. For sure in others the situation is different, but the present manuscript appears to suggest that cytoplasmic streaming is always associated with interfilament sliding, whereas that is not the case. This needs to be clarified.

The manuscript contains various measurements and predictions, but it is rather unclear which of them would unambiguously demonstrate the postulated mechanism if recovered in experiments. This point is crucial and should come across the manuscript with no ambiguity. In other words, the authors should clearly explain which one of their predictions an experimentalist should recover in order to verify that the mechanism behind MTs streaming is indeed that proposed in this paper.

There is a significant literature on the streaming problem in *Drosophila* that has not been cited. Examples include Woodhouse, et al., 2013 and Khuc Trong, et al. 2015, which discuss in detail self-organization processes and the role of cytoskeletal architecture on streaming patterns. Of particular interest in the case of *Drosophila* is the nucleation of microtubules from the periphery of the oocyte, leading to anchoring there. This is not accounted for in the present paper.

2) The authors do not explain what promotes short-ranged polar alignment in the absence of directed motion and cross-linking. In other words, why the polar bundles form in the first place? Can the authors exclude that the interlocking of the beads forming an individual MT has nothing to do with it?

3) While reading the manuscript, it is very tempting to think about the active suspensions of MT bundles and kinesin pioneered in the Dogic Lab. The authors do refer to some of this work and, toward the end of the manuscript, explicitly say that the collective motion found here resembles that observed in those experiments. Yet, it is rather puzzling that they avoid making a direct comparison. If the present numerical approach could serve as a particle model of these active suspensions, this should be clearly said and motivated (with an eye to the rich theoretical literature around the topic). If not, it would be useful to know where are the fundamental differences and how both these model systems compare to cytoplasmatic streaming in vivo.

4) It is somewhat unclear how the model outlined in Sec. II. reconciles with the non-equilibrium nature of kinesin-based propulsion. MT-kinesin interactions are modeled through conservative forces, that can be expressed as derivatives of of potential energy U_mot. Furthermore, the motor binding rate follows the Boltzmann distribution. This raises the question of whether the authors are attempting to describe cytoskeletal activity as an equilibrium process. Most models of cytoskeletal fluids (both discrete and continuous) are based on the assumption that kinesin moves at constant speed from the minus to the plus end (or vice versa for specific types of kinesin). This manifestly violates detailed balance and is consistent with experimental observations (see e.g. Schnitzer and Block, Nature 1997). One can debate on whether kinesin is in fact delivering a constant power, as opposed to move at constant speed, but both scenarios appear to lie outside of the scope of the present model.

5) The reviewers raised questions about the degree of novelty of your methodology. While recognizing that the specific results in the paper concerning sliding motility are new, they pointed to recent work published by the Shelley group (Nazockdast et al., 2017) which has introduced a 3D computational framework that accounts for polymerization and depolymerization kinetics of fibers, their interactions with molecular motors and other objects, their flexibility, and hydrodynamic coupling. Their model has been applied in (Nazockdast et al., 2017).

The present authors should clearly compare and contrast their work with these recent papers.

6) Regarding the simulations, the reviewers are unclear why a Langevin equation with a mass term was used in what is clearly an overdamped problem. It is also unclear why it is possible a priori to neglect hydrodynamic interactions between filaments, and the significance of working in only two dimensions. All of

these issues need clarification.

Reviewer #1:

This article describes a computational study of a model for cytoplasmic streaming, motivated by the phenomenology found in *Drosophila* oocytes. The authors introduce a coarse-grained model in which details of the molecular motors' activity that can lead to sliding between adjacent MTs are subsumed into an effective orientation-dependent potential. The MTs are modelled as linked spheres, and the whole setup is described by a Langevin equation. The results include many aspects of the correlation functions between the microtubules, with different regimes characterized by different types of alignments. Probability distribution functions of speeds are computed, of the kind that can be measured experimentally. The authors make some contact with experimental studies of streaming using PIV and advance the hypothesis that heavy-tailed velocity distributions arise without the previously conjectured need for varying motor speeds or complex cytoskeletal geometry.

The subject matter of this paper is certainly appropriate for *eLife*, and as a computational study it is reasonably well done. Less clear to me is the significance of the results. In part this is due to what appears to be a superficial understanding of the literature on streaming. Streaming occurs in a vast range of systems, from plants upwards, and in many of these systems the filaments are both organized and unchanging in their conformations. For sure in others the situation is different, but the present manuscript appears to suggest that cytoplasmic streaming is always associated with interfilament sliding.

Second, there is a significant literature on the streaming problem in *Drosophila* that has not been cited. Examples include Woodhouse et al., 2013 and Khuc Trong, et al. 2015, which discuss in detail self-organization processes and the role of cytoskeletal architecture on streaming patterns. Of particular interest in the case of *Drosophila* is the nucleation of microtubules from the periphery of the oocyte, leading to anchoring there. This is not accounted for in the present paper.

Regarding the simulations, I am unclear why the authors would solve a Langevin equation with a mass term in what is clearly an overdamped problem. It is also unclear to me why it is possible a priori to neglect hydrodynamic interactions between filaments, and the significance of working in two dimensions.

Overall, I think the contributions of this paper are interesting, but the lack of proper biological context is a significant weakness.

Reviewer #2:

The manuscript by Ravichandran et al. introduces a computational framework for studying microtubule (MT) dynamics with focus on motor-driven sliding motility. MTs are modeled as spring chains with standard stretching and bending energy contributions, and steric MT-MT interactions are described by WCA repulsion. Motor activity is modeled by cross-linker springs between MTs that bind with exponential rates depending on relative orientation between motors and MT pairs. The model neglects hydrodynamics and does not account for MT nucleation and de/polymerization, although both effects could likely be added in future extensions of this framework. Simulations are restricted to 2D systems. Simulated MT numbers are O(1000) and the authors make a commendable effort to provide biologically relevant values for all model parameters.

The paper is clearly written and the numerical study has been performed carefully.

My main concern regarding suitability for publication in *eLife* is novelty.

Recent work published by the Shelley group, see Nazockdast et al., 2017, has introduced a 3D computational framework that accounts for polymerization and depolymerization kinetics of fibers, their interactions with molecular motors and other objects, their flexibility, and hydrodynamic coupling. Their model has been applied in Nazockdast et al., 2017.

In view of this previously published work, I believe that the present manuscript does not constitute the type of major conceptual or computational advance typically expected for publication in *eLife*. That said, it seems to me that the specific results in the paper concerning sliding motility are new and certainly deserve publication in some other form.

Reviewer #3:

Ravichandran and coworkers report a comprehensive computational study of microtubules (MT) streaming. This process, observed both in vivo and in vitro, is generally ascribed to the sliding motion promoted by kinesin molecules, but the microscopic mechanism behind the kinesin-mediated MT-MT interactions is still debated. Numerical simulations suggest that streaming results from the interaction between bundles of polar-aligned MTs and is particularly sensitive to the time scale associated with the reorientation of individual MTs. The paper appears technically sound, clearly written and nicely illustrated. Unfortunately, there are various points where the authors have been too vague.

1) Whereas the mechanism postulated by authors does produces streaming in the in silico system studied here, it is not guaranteed that the same mechanism occurs inside the cell or in vitro. The manuscript contains various measurements and predictions, but it is rather unclear which of them would unambiguously demonstrate the postulated mechanism if recovered in experiments. This point is crucial in my opinion and should come across the manuscript with no ambiguity. In other words, the authors should clearly explain which one of their predictions should an experimentalist recover in order to verify that the mechanism behind MTs streaming is indeed that proposed in this paper.

2) The authors do not explain what promotes short-ranged polar alignment in the absence of directed motion and cross-linking. In other words, why the polar bundles form in the first place? Can the authors exclude that the interlocking of the beads forming an individual MT has nothing to do with it?

3) While reading the manuscript, it is very tempting to think about the active suspensions of MT bundles and kinesin pioneered in the Lab of Zvonimir Dogic and now investigated by various other groups around the world. The authors do refer to some paper by the Dogic Lab and, toward the end of the manuscript, explicitly say that the collective motion found here resembles that observed in those experiments. Yet, they avoid making a direct comparison. I find this puzzling. If the present numerical approach could serve as a particle model of Dogic's active suspensions, this should be clearly said and motivated (with an eye to the rich theoretical literature around the topic). If not, it would be useful to know where are the fundamental differences and how both these model systems compare to cytoplasmatic streaming in vivo.

4) It is somewhat unclear how the model outlined in Sec. II. reconciles with the non-equilibrium nature of kinesin-based propulsion. MT-kinesin interactions are modeled through conservative forces, that can be expressed as derivatives of of potential energy U_mot. Furthermore, the motor binding rate follows the Boltzmann distribution. This raises question on whether the authors are attempting to describe cytoskeletal activity as an equilibrium process. Most of models of cytoskeletal fluids (both discrete and continuous) are based on the assumption that kinesin moves at constant speed from the minus to the plus end (or vice versa for specific types of kinesin). This manifestly violates detailed balance and is consistent with experimental observations (see e.g. Schnitzer and Block, Nature 1997). One can debate on whether kinesin is in fact delivering a constant power, as opposed to move at constant speed, but both scenarios appear to lie outside of the scope of the present model.

In summary, whereas the authors have been quite meticulous in calibrating the parameters to experimental values, it is unclear to me whether what they present are indeed properties of MTs and kinesin or simply properties of their model. Therefore, I am unable to recommend this paper for publication in *eLife* in the present form.

---

## [Author Response]

[Editors' note: the authors’ plan for revisions was approved and the authors made a formal revised submission.]

Essential revisions:1) Whereas the mechanism postulated by authors does produces streaming in the in silico system studied here, it is not guaranteed that the same mechanism occurs inside the cell or in vitro. Streaming occurs in a vast range of systems, from plants upwards, and in many of these systems the filaments are both organized and unchanging in their conformations. For sure in others the situation is different, but the present manuscript appears to suggest that cytoplasmic streaming is always associated with interfilament sliding, whereas that is not the case. This needs to be clarified.The manuscript contains various measurements and predictions, but it is rather unclear which of them would unambiguously demonstrate the postulated mechanism if recovered in experiments. This point is crucial and should come across the manuscript with no ambiguity. In other words, the authors should clearly explain which one of their predictions an experimentalist should recover in order to verify that the mechanism behind MTs streaming is indeed that proposed in this paper.There is a significant literature on the streaming problem in Drosophila that has not been cited. Examples include Woodhouse, et al., 2013, and Khuc Trong, et al. 2015, which discuss in detail self-organization processes and the role of cytoskeletal architecture on streaming patterns. Of particular interest in the case of Drosophila is the nucleation of microtubules from the periphery of the oocyte, leading to anchoring there. This is not accounted for in the present paper.

We acknowledge that cytoskeletal streaming has been reported across multiple biological systems and that various mechanisms can account for this motion. Our goal is to isolate and address anti-aligned filament sliding as a relevant mechanism of activity, and to study whether it alone could be responsible for streaming. Through our findings, we wish to emphasize that this mechanism can singly bring about some of the hallmarks of MT streaming observed in experiments; however, it is largely underappreciated in both biological and soft-matter literature so far. We thus propose that further theoretical and experimental studies of filament sliding and MT streaming are required to exactly determine their role in long-range MT organization and transport in various contexts (e.g. neurons, *Drosophila* oocytes). We certainly do not want to claim that filament sliding is only and always responsible for MT streaming.

We agree with the reviewers that this point is not clearly discussed in our manuscript. We now state this more clearly in the Introduction by discussing other “bottom-up” in vitro and *in silico* approaches and contrasting them with “top-down” approaches. For the latter, we discuss other streaming mechanisms, such as the cargo transport reported for *Chara corallina* by Woodhouse et al., 2013, and nucleation of MTs from the periphery reported for *Drosophila* oocytes by Khuc Trong et al. 2015.

An important point is that we focus in our study on the motion on several scales far from boundaries. We have extended a paragraph in the Discussion section on how our model can be tested by experimentalists. We think here mainly about fluorescence experiments that compare both, collective and single-filament dynamics. Addition of small fractions of fluorescent filaments should allow the analysis of single-filament motion (Levy flight-like motion, velocity distributions), while photobleaching or photoactivation experiments can reveal collective motion.

2) The authors do not explain what promotes short-ranged polar alignment in the absence of directed motion and cross-linking. In other words, why the polar bundles form in the first place? Can the authors exclude that the interlocking of the beads forming an individual MT has nothing to do with it?

We believe that this is a misunderstanding regarding the non-equilibrium nature of our simulations, see also point 4) of the essential revisions. In absence of directed motion (p_a_=0, thermal equilibrium) polar alignment cannot be obtained in our simulations. For suspensions of passive filaments, neighboring filaments do not experience sliding forces and therefore nematic and polar configurations are indistinguishable. For suspensions of active filaments, a filament in an oppositely-oriented region is propelled and transported after several simulation steps to an aligned region, in which case the active propulsion ceases.

Independent of polar alignment, potential interlocking of rough filaments is indeed an important issue to be considered carefully in the simulations. In previous work reported by Abkenar et al., 2013, Isele-Holder et al., 2015, Abaurrea Velasco et al. (Soft Matter 13, 5865 (2017)), and Duman et al., 2018, we have studied the effect of filament discretization on the dynamics of active systems. We found that by using overlapping beads within filaments, where the bond-length is half the cut-off of the excluded volume potential, the friction is strongly reduced. In particular the filament model in Isele-Holder et al. is exactly the same as the model used in this manuscript, which we now state in section Materials and methods, System parameters.

3) While reading the manuscript, it is very tempting to think about the active suspensions of MT bundles and kinesin pioneered in the Dogic Lab. The authors do refer to some of this work and, toward the end of the manuscript, explicitly say that the collective motion found here resembles that observed in those experiments. Yet, it is rather puzzling that they avoid making a direct comparison. If the present numerical approach could serve as a particle model of these active suspensions, this should be clearly said and motivated (with an eye to the rich theoretical literature around the topic). If not, it would be useful to know where are the fundamental differences and how both these model systems compare to cytoplasmatic streaming in vivo.

We thank the reviewers for raising the issue of the connection between our work and the active-suspension model systems consisting of MT bundles and kinesin motors pioneered in the Dogic lab. The work of Dogic et al. has indeed been a very important motivation for us, and we completely agree that a more detailed comparison is desirable. Dogic’s studies of 2D model systems -- with MTs and kinesins at an oil-water interface – do not mimic or reproduce any specific biological cell in vivo but instead aim at a general understanding of self-organization and dynamics of MTs by the activity generated by molecular motors.

In the previous version of the manuscript, we referred to Dogic‘s active suspensions only for very specific results in the Introduction section. In the Discussion section, we referred to the hierarchical nature of the 2D systems. The main difference between both studies is a mismatch in system size between the experimental model system and our theoretical / numerical model. While the single-filament resolution of our simulations is not achieved in the experiments, the simulations deal with much smaller systems than the experiments. Furthermore, the two model systems are different regarding the filament aspect ratio (6:1) and the filament length to system size ratio. Filament and motor concentrations at the interface are unknown in the experiments and the filaments bundle. Therefore, a comparison of results for the two systems is at this stage only possible on a qualitative level.

We have added a new paragraph on in vitro and *in silico* approaches to the Introduction, where we introduce the active suspensions pioneered in the Dogic lab as well as computer simulations and where we highlight the single-filament vs. bundle nature. We contrast these “bottom-up” model systems with “top-down” studies for cytoplasmic streaming in vivoin the following paragraph. See also our reply to point 1).

We have also added a new paragraph to the Discussion section and Figure 3—figure supplement 10 that compares our results with Dogic’s results. Here, we focus on mean squared displacements.

In Figure 3—figure supplement 10, we compare the MSDs from our simulations to those measured experimentally by Dogic et al., in 2012. The MSDs have been measured using micron sized tracer particles immersed in the active nematic in Dogic’s work and using individual filaments in our work. The different system parameters make it difficult to find a common normalization of the lag time as well as of the MSDs. Following the concept of our manuscript, we have normalized time by the single-filament rotation time and the MSD by the filament length squared. Despite the difficulties to obtain a quantitative comparison, the qualitative features of the MSDs are very similar. For passive systems, the MSDs represent diffusive motion. When the motor probability p_a_ in our model or the ATP concentration in the experiments is increased, the filaments become more active and the active velocity leads in both cases to an active ballistic motion. Our simulations, however, show in addition also very clearly the active diffusive regime at long times (the plateau). Here the filaments lose their orientational correlation. In contrast, in Dogic’s systems the filaments move mostly on straight trajectories for all studied lag times. The (active) rotational diffusion in our simulations is enhanced because of a lower filament density and a smaller MT aspect ratio compared with the experiments. The shorter active rotation time is also the reason why the active ballistic regime (MSD~(vt)^2^) is not fully developed in our simulations, but affected by crossover regimes to diffusive motion.

4) It is somewhat unclear how the model outlined in Sec. II. reconciles with the non-equilibrium nature of kinesin-based propulsion. MT-kinesin interactions are modeled through conservative forces, that can be expressed as derivatives of of potential energy U_mot. Furthermore, the motor binding rate follows the Boltzmann distribution. This raises the question of whether the authors are attempting to describe cytoskeletal activity as an equilibrium process. Most models of cytoskeletal fluids (both discrete and continuous) are based on the assumption that kinesin moves at constant speed from the minus to the plus end (or vice versa for specific types of kinesin). This manifestly violates detailed balance and is consistent with experimental observations (see e.g. Schnitzer and Block, Nature 1997). One can debate on whether kinesin is in fact delivering a constant power, as opposed to move at constant speed, but both scenarios appear to lie outside of the scope of the present model.

We thank the reviewers for raising the issue that the current presentation of the model may be confusing regarding the implementation of the active motor force. The nonequilibrium activity is a key aspect of our model, and it fully captures the non-equilibrium nature of the MT-kinesin systems. The activity enters via a periodic switching on and off of harmonic potentials that are generated by the extension of the molecular motors – very similar to a ratchet model. Here, the requirement of acute angles between the stalks of the motors and the polar filaments generates net propulsion forces. To better highlight the non-equilibrium nature, we have added a new paragraph on the nonequilibrium nature of the system to the general part of the Introduction. In addition, we have added a short statement before Equation 6, which emphasizes that because of the temporary and orientation-dependent “motor bonds” the system is inherently out of equilibrium.

5) The reviewers raised questions about the degree of novelty of your methodology. While recognizing that the specific results in the paper concerning sliding motility are new, they pointed to recent work published by the Shelley group (Nazockdast, et al., 2017) which has introduced a 3D computational framework that accounts for polymerization and depolymerization kinetics of fibers, their interactions with molecular motors and other objects, their flexibility, and hydrodynamic coupling. Their model has been applied in Nazockdast, et al., 2017. The present authors should clearly compare and contrast their work with these recent papers.

The model of the Shelley group features several important aspects of cytoskeletal systems including filament flexibility, filament polymerization and depolymerization, and hydrodynamic interactions. Steric interactions between filaments are not included. Consequently, the authors have studied a system where steric interactions are not expected to play a major role, the mitotic positioning where the microtubules are oriented radially. Our model features filament flexibility, an effective-motor potential between neighboring antiparallel filaments that coarse-grains walking of the motors, and steric interactions between the filaments. Hydrodynamic interactions are not included. We study dense systems, where the screening of hydrodynamic interactions may decrease their importance and at the same time steric interactions are essential. Both simulation frameworks use parallelized codes to allow for investigation of large systems. Our code builds on the freely available LAMMPS simulation package. Together with the source code and the input file provided with this submission, the simulations can easily be implemented by other researchers.

We have added a new paragraph to the section Introduction, Coarse-grained model, where we compare and contrast our work with the model the Shelley group (New York) and the Cytosim simulation package of the Nedelec group (Heidelberg).

In particular, we also highlight that our filament-based simulations are dissimilar from calculations and simulations that strive to capture microscopic, biological details. We coarse-grain individual motor interactions into an anti-aligned motor potential, and we show that our model still captures hallmarks of MT streaming. A novel aspect is that we demonstrate that microscopic details of motor interaction might be unnecessary when trying to capture long length- and time-scale phenomena. Moreover, we distill processes occurring in various time scales into observables which can be identified and verified through other simulations and experiments.

6) Regarding the simulations, the reviewers are unclear why a Langevin equation with a mass term was used in what is clearly an overdamped problem. It is also unclear why it is possible a priori to neglect hydrodynamic interactions between filaments, and the significance of working in only two dimensions. All of these issues need clarification.

The referees are correct that the dynamics in the experimental system is overdamped. We use Langevin Dynamics (LD) simulations with a mass term instead of a Brownian Dynamics (BD) simulation because of the numerical stability of the Langevin approach. When the mass term is included, up to 100 times longer time steps can be used in the simulations. However, we operate the code with parameters that ensure that the systems are in the overdamped regime for the experimentally relevant time scales.

**Author response image 1. respfig1:** Mean squared displacements of filaments obtained from Langevin Dynamics (LD) and Brownian Dynamics (BD), for single filaments (left) and filaments in suspensions with packing fraction 𝜙 = 0.3 (right). The simulation parameters are the same as in the manuscript.

Author response image 1 shows a comparison of LD and BD simulations for (a) single filaments and (b) dense systems. The results demonstrate that our choice of parameters is justified because the crossover time from (inertial) ballistic to diffusive motion is at much shorter times than the time for a filament to diffuse over its own length. We have added a short paragraph to the section Materials and methods, Langevin Dynamics.

Hydrodynamic interactions between filaments may very well play an important role for cytoplasmic streaming. Nevertheless, there are several reasons why it makes sense to study a model in which hydrodynamic interactions are not taken into account:

1) Hydrodynamic interactions are long-ranged, and therefore induce coupling of motion not only of neighboring but also of distant filaments. Furthermore, they require numerically a significantly larger effort, as well as large system sizes. Simulations that include hydrodynamic interactions are therefore usually restricted to smaller number of particles.

2) In order to elucidate mechanisms, it is often helpful to start from simple systems, and to add additional features subsequently.

3) Boundary effect can strongly affect hydrodynamic flows, and hydrodynamic interactions are screened near surfaces and membranes.

4) In fact, our model for the filament-motor systems can be combined rather easily with particle-based mesoscale hydrodynamic approaches, which have been developed in recent years, and which are ideally suited for this purpose. Two examples for previous studies of filament hydrodynamics in a different context are:

"Semidilute polymer solutions at equilibrium and under shear flow", C.-C. Huang, R.G. Winkler, G. Sutmann, and G. Gompper, Macromolecules 43, 10107 (2010);

“Migration of semiflexible polymers in microcapillary flow", R. Chelakkot, R.G. Winkler, and G. Gompper, EPL 91, 14001 (2010).

5) It will certainly be very interesting to investigate the effect of hydrodynamic interactions in our model, which we plan to do in the near future. However, this goes far beyond the current study.

We have studied 2D systems motivated by Dogic’s model systems for active suspensions. For these experimental model systems, the dynamics of the entire systems, i.e., of all filaments, can be followed using light microscopy. Furthermore, the 2D geometry might be relevant for filament-motor systems close to boundaries, such as for cytoskeletal filaments next to plasma membranes of cells. We now motivate the reduced dimensionality in the paragraph on in vitro and in silico models in the Introduction, where we highlight that such less complex bottom-up studies are especially suited to study specific mechanisms in more detail.